# ATTENTION IN LARGE LANGUAGE MODELS YIELDS EFFICIENT ZERO-SHOT RE-RANKERS

**Shijie Chen**  **Bernal Jiménez Gutiérrez**  **Yu Su**
The Ohio State University
{chen.10216,jimenezgutierrez.1,su.809}@osu.edu

## ABSTRACT

Information retrieval (IR) systems have played a vital role in modern digital life and have cemented their continued usefulness in this new era of generative AI via retrieval-augmented generation. With strong language processing capabilities and remarkable versatility, large language models (LLMs) have become popular choices for zero-shot re-ranking in IR systems. So far, LLM-based re-ranking methods rely on strong generative capabilities, which restricts their use to either specialized or powerful proprietary models. Given these restrictions, we ask: *is autoregressive generation necessary and optimal for LLMs to perform re-ranking?* We hypothesize that there are abundant signals relevant to re-ranking within LLMs that might not be used to their full potential via generation. To more directly leverage such signals, we propose *in-context re-ranking* (ICR), a novel method that leverages the change in attention pattern caused by the search query for accurate and efficient re-ranking. We assume that more relevant documents should receive more attention weights when an LLM is processing the query tokens, and leverage such signals for re-ranking. To mitigate the intrinsic biases in LLMs, we propose a calibration method using a content-free query. Due to the absence of generation, ICR only requires two ($O(1)$) forward passes to re-rank $N$ documents, making it substantially more efficient than generative re-ranking methods that require at least $O(N)$ forward passes. Our novel design also enables ICR to be applied to any LLM without specialized training while guaranteeing a well-formed ranking. Extensive experiments with two popular open-weight LLMs on standard single-hop and multi-hop information retrieval benchmarks show that ICR outperforms RankGPT while cutting the latency by more than 60% in practice. Through detailed analyses, we show that ICR's performance is specially strong on tasks that require more complex re-ranking signals, such as handling *contextualization* and *contradiction* between the query and passages, as well as *information integration* across multiple passages. Our findings call for further exploration on novel ways of utilizing open-weight LLMs beyond text generation. [1]

## 1 INTRODUCTION

Information retrieval (IR) plays a pivotal role in numerous real-world applications, ranging from search engines to recommendation systems. In the era of generative AI, IR techniques are widely used to enhance generative models with up-to-date information, a process known as retrieval-augmented generation (Lewis et al., 2020). A critical component of modern IR pipelines is re-ranking (Nogueira et al., 2019; 2020), which typically leverages more powerful models to improve search outcomes produced by lightweight, large-scale retrieval systems such as BM25 (Robertson & Zaragoza, 2009) and dense retrieval systems (Karpukhin et al., 2020). The advent of Large Language Models (LLMs) has significantly influenced the technology stack of information retrieval (Zhu et al., 2023; Wang et al., 2024), particularly in their application as high-performance zero-shot re-rankers, leading to substantial improvements in retrieval performance (Sun et al., 2023).

Existing LLM-based re-ranking methods predominantly rely on the generative capabilities of LLMs. Listwise methods (Sun et al., 2023; Tang et al., 2023; Pradeep et al., 2023b) requires the LLM to

---

[1] Code and data: https://github.com/OSU-NLP-Group/In-Context-Reranking.

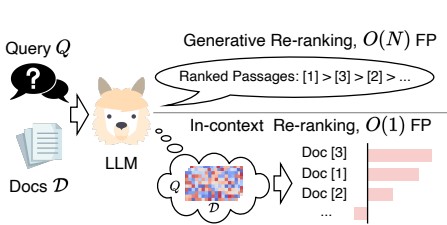

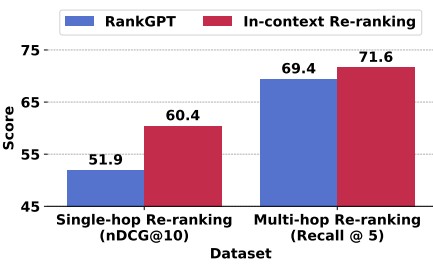

(a) LLM-based re-ranking methods. FP: forward pass.      (b) Re-ranking performance of Llama 3.1 8B.

Figure 1: (a) Overview of in-context re-ranking (ICR). (b) Re-ranking performance using Llama-3.1 8B on single-hop and multi-hop re-ranking tasks. We report the micro-average across tasks.

generate a ranking of document identifiers in a pre-defined format. Comparison-based methods, such as pointwise(Zhuang et al., 2023a;b; Reddy et al., 2024) and pairwise methods(Qin et al., 2024; Chen et al., 2024), either ask the LLM to generate a relevance score for each document separately, or to compare document pairs and form a ranking. Other methods utilize the LLM's output logits to rank document identifiers and avoid expensive generation(Zhuang et al., 2024; Reddy et al., 2024). So far, LLM-based re-rankers depend on strong generative capabilities, and thus have only proven to be effective when using strong proprietary models or supervised fine-tuned models as the backbone, constraining their broader adoption with open-weight models.

Considering the limitations of existing generative approaches, we ask: *Is autoregressive generation necessary and optimal for LLMs to perform re-ranking?* To answer this research question, we hypothesize that abundant signals related to re-ranking have already emerged during the context encoding stage of LLMs and directly utilizing such signals may improve re-ranking performance over using them indirectly via generative approaches. In this work, we propose *in-context re-ranking* (ICR), a simple yet effective re-ranking method that harnesses the contextual understanding capacity of LLMs through their attention weights, as they correlate closely with context relevance within LLMs (Peysakhovich & Lerer, 2023). ICR first aggregates the attention weights received by each document token from all query tokens. To mitigate intrinsic biases of LLMs, we calibrate the ranking scores by subtracting the attention scores obtained by using a content-free query (i.e. "N/A"). Finally, we take the sum of the calibrated attention weights from all tokens in a document as the final ranking score. In this way, the ranking score captures the change in total attention received by each document caused by the semantics of the actual query. As shown in Figure 1, ICR only requires two forward passes to re-rank $N$ documents, substantially reducing the $O(N)$ forward passes required by generative re-ranking (Qin et al., 2024) to $O(1)$. Additionally, without the need of text generation, ICR guarantees a well-formed ranking using any LLM. The token-level ranking scores also provide improved interpretability compared to ranking results generated in black-box processes.

We comprehensively evaluate ICR with two open-weight LLMs and various IR benchmarks, including the well-established TREC (Craswell et al., 2020) and BEIR benchmark (Thakur et al., 2021), as well as multi-hop retrieval tasks (Gutiérrez et al., 2024). Our experiments show that ICR outperforms RankGPT on most tasks while reducing latency by more than 60%. When used with Llama-3.1 8B (Dubey et al., 2024), ICR outperforms RankGPT (Sun et al., 2023) by $8.4$ points in single-hop re-ranking and $2.2$ points in multi-hop re-ranking on average, closely matching the strong proprietary model GPT-4o mini(OpenAI, 2024).

Through detailed analysis, we demonstrate that LLMs' attention weights capture rich signals relevant to re-ranking, including *contextualization signals* between query and passages, *reasoning signals* for handling contradiction relations; and *information integration signals* involving bridge entities. During generation, such signals may be overpowered by others, such as position bias and lexical overlap. Without generation, ICR is able to directly leverage these relevance signals, which substantially improves re-ranking performance, especially on tasks that require handling deeper semantic relevance between query and documents, such as evidence retrieval for fact verification, as well as tasks that require integrating information from multiple documents, as is the case with multi-hop re-ranking tasks. Our findings indicate that generation is not always the optimal way of unleashing the full potential of LLMs, and highlights the value of exploring creative ways to better leverage open-weight models across diverse applications.

## 2 RELATED WORK

### 2.1 ZERO-SHOT RE-RANKING WITH LARGE LANGUAGE MODELS

Existing LLM-based re-ranking methods typically fall into three categories: pointwise, pairwise, and listwise re-ranking. Pointwise re-ranking asks the LLM to grade the relevance of each document separately. This can be done via relevance generation (Liang et al., 2023) and query generation (Drozdov et al., 2023; Sachan et al., 2022). However, the black-box nature of LLMs makes calibrating the pointwise scores an open challenge. Recently, Qin et al. (2024) propose a pairwise re-ranking method that first uses LLMs for pairwise document comparisons and then integrates the results into a complete ranking. A more direct approach is listwise re-ranking (Sun et al., 2023; Ma et al., 2023) which uses the LLM to generate a ranking list. As pointed out by Qin et al. (2024), all these methods require $O(N)$ to $O(N^2)$ API calls for re-ranking $N$ documents. Since each API call generates at least $O(1)$ tokens, their complexity measured in forward passes remains at least $O(N)$. Please see Appendix A for a more detailed complexity analysis. Apart from cost concerns, two additional issues stand out for generative re-ranking: (1) the relevance scores generated by LLMs lack interpretability, which undermines the trustworthiness of the re-ranking process; (2) there is no guarantee that LLMs always generate well-formed outputs, such as relevance scores or ranking lists. This greatly limits the availability of LLM-based re-rankers and the usage of language models that are weaker at following instructions.

To enable zero-shot re-ranking with open-weight LLMs, researchers also specialize foundation language models with re-ranking data synthesized by strong proprietary LLMs (Sun et al., 2023; Pradeep et al., 2023a;b). Nevertheless, they have been shown to struggle with generalizing to corpus sizes unseen during training (Pradeep et al., 2023b). More recently, FIRST (Reddy et al., 2024) achieves zero-shot re-ranking with one forward pass by sorting document identifiers based on the output logits of the first generated token, which is supposed to represent the most relevant document. However, FIRST still requires specialized fine-tuning, and its logit-based design restricts document identifiers to be single-token (i.e. "A" to "Z"), limiting the number of documents it can handle .

In contrast to generative re-ranking methods, ICR is based on the general language processing abilities present in any pretrained language model. Without generation, ICR avoids the instability of generation, guaranteeing well-formed and consistent ranking results. As a result, ICR is applicable to any LLM without specialized fine-tuning. Additionally, due to the limitation of context length and the degradation of generation quality for long contexts, existing zero-shot methods typically need to be used with a sliding window mechanism (Sun et al., 2023), which requires $O(N)$ iterations, to handle larger retrieval corpora. In comparison, ICR can fully leverage recent long-context LLMs and re-ranks any number of documents with two ($O(1)$) forward passes, substantially lowering latency.

### 2.2 ATTENTION-BASED INTERPRETABILITY IN LARGE LANGUAGE MODELS

Attention weights has been used for interpreting the importance of input tokens to language model output since the attention mechanism was first introduced by Bahdanau et al. (2015). While there exist controversial opinions on how reliable attention weights are for interpretability (Serrano & Smith, 2019; Wiegreffe & Pinter, 2019), previous work has shown the effectiveness of distilling cross-attention between documents and a reader model for training better retrievers (Izacard & Grave, 2021). More recently, Peysakhovich & Lerer (2023) propose attention sorting to improve generation quality, where input documents are sorted by the amount of attention they receive when the LLM is generating the next token. In-context re-ranking shares a similar assumption that attention weights correlates with the relevance of documents to the query. However, we aggregate the attention weights over all query tokens, rather than only considering the last token as in previous work. We also propose a calibration method to help mitigate LLMs' intrinsic biases. Our experiments on information retrieval benchmarks show the necessity of the proposed components.

## 3 IN-CONTEXT RE-RANKING

In this section, we introduce the in-context re-ranking (ICR) method. Given a query $Q$ and a set of $N$ documents $\mathcal{D} = \{d_1, \ldots, d_N\}$ retrieved by a retriever, a re-ranker ranks the documents by their relevance to the query. ICR accomplishes the re-ranking process by producing a ranking score

Figure 2: Illustration for In-context Re-ranking (ICR). ICR first aggregates (Agg) attention weights between document and query tokens to form token-level query ranking scores. Then a calibration score is calculated similarly with the calibration query, which is subtracted form the query ranking score. The final score is the sum of calibrated scores for all tokens for each document.

$s_i$ for each document $d_i$ using the attention distribution within an LLM. More specifically, we first prompt the LLM with the documents and the search query to retrieve attention weights (Section 3.1). Then we aggregate the attention weights across all attention heads and layers to compute a token-level ranking score for each document (Section 3.2). Finally, we calibrate the ranking scores with a content-free query (Section 3.3). Figure 2 illustrates the architecture of ICR.

## 3.1 PROMPT DESIGN

To leverage the instruction following abilities in modern LLMs, we formulate the re-ranking task into either question answering (QA) or information extraction (IE) tasks, which are more commonly seen by LLMs. We design two kinds of instruction prompts for different type of queries:

**QA style:** For search queries that are questions, we prompt the LLM to answer the question based on provided documents, leveraging its QA abilities.

**IE style:** For non-question queries, we prompt the LLM to find relevant information from provided documents, leveraging its IE abilities.

The prompt consists of the instruction, the documents to be re-ranked, and the search query. We reverse the order of input documents as we find it improves performance in our preliminary experiments, probably due to the position bias in LLMs. Given the autoregressive nature of decoder-only LLMs, we place the query at the end of the prompt. Please see Appendix B for prompt examples.

## 3.2 ATTENTION AGGREGATION

We hypothesize that LLMs will, on average, attend more strongly to tokens in relevant documents than irrelevant ones when processing the query. Accordingly, we design an attention aggregation method to compute the attention weight that each token in a document gets from the query across all attention heads and layers in the LLM. Using an LLM with $L$ layers and $H$ attention heads, the ranking score $s_{d_{i,j}}$ for the $j$-th token in document $d_i$ is:

$$s_{d_{i,j},Q} = \frac{1}{|\mathcal{I}_Q|} \sum_{l=1}^{L} \sum_{h=1}^{H} \sum_{k \in \mathcal{I}_Q} a_{j,k}^{l,h}, \tag{1}$$

where $\mathcal{I}_Q$ represents the set of token indices for query $Q$. $a_{j,k}^{l,h}$ denotes the attention weight from the $k$-th token (in the query) to the $j$-th token (in document $d_i$) by the $h$-th attention head at layer $l$. Since each query token's attention weights form a distribution and sum up to 1, our aggregation strategy avoids the commonly seen length bias issue where longer documents receive higher scores.

## 3.3 RANKING SCORE CALIBRATION

LLMs show various intrinsic biases in text generation (Gallegos et al., 2024). An ideal re-ranker should assign equal scores to all documents when given a content-free query. Inspired by Zhao et al. (2021), we propose a ranking score calibration method to mitigate intrinsic biases in LLMs' attention weights. Following Zhao et al. (2021), we use *"N/A"* as the calibration query $Q_{cal}$ and calculate

calibration scores $\{s_{d_{i,j},Q_{cal}}\}$ via attention aggregation for each document. Then we subtract it from ranking scores from the actual query $\{s_{d_{i,j},Q}\}$ to obtain the calibrated ranking score $\{s_{d_{i,j}}\}$:

$$s_{d_{i,j}} = s_{d_{i,j},Q} - s_{d_{i,j},Q_{cal}}. \tag{2}$$

As will be discussed in Section 5.1, the calibration scores capture strong attention weights biased towards meaningless tokens in the documents, such as punctuation, which should not affect the relevance of the documents. After that, we filter out such tokens with abnormally negative calibrated scores. Finally, we sum the calibrated scores for all tokens in each document to obtain the final ranking score $s_{d_i}$, which measures the change of attention weights that each document receives when the query changes from the content-free calibration query to the actual query:

$$
\begin{aligned}
\mathcal{S}_{d_i} &= \{d_{i,j}\}, \\
\mathcal{S}_{d_i}^* &= \{d_{i,j} | d_{i,j} > \bar{\mathcal{S}}_{d_i} - 2\sigma_{\mathcal{S}_{d_i}}\}, \\
s_{d_i} &= \sum_{s \in \mathcal{S}_{d_i}^*} s,
\end{aligned}
\tag{3}
$$

where $\sigma$ denotes standard deviation. Since we place the query tokens at the end of the re-ranking prompt, ICR can share the KV cache of document tokens when computing $s_{d_i,Q}$ and $s_{d_i,Q_{cal}}$. Empirically, we observe the overhead for calibration to be only approximately 30%.

## 4 EXPERIMENTS

We evaluate the proposed in-context re-ranking method on both single-hop (Section 4.2) and multi-hop (Section 4.3) re-ranking tasks using open-weight LLMs. Considering the additional training cost introduced by supervised methods, such as RankVicuna (Pradeep et al., 2023a), and inference cost needed by other listwise methods, such as the setwise method (Zhuang et al., 2024), we mainly use RankGPT as a representative zero-shot baseline (Comparisons with those two methods are in Appendix F.). We also examine the scaling trend of their efficiency and performance (Section 4.4). Then we show the effectiveness of attention aggregation and calibration in ICR via ablation studies (Section 4.5). As specialized methods do not outperform their teacher models (Pradeep et al., 2023b; Reddy et al., 2024), RankGPT with strong proprietary models still strikes the best balance between performance and efficiency for zero-shot re-ranking. We report re-ranking results from RankGPT with GPT3.5-turbo and GPT-4o mini for reference.

### 4.1 EXPERIMENT SETUP

**Base LLMs.** As in-context re-ranking requires access to the attention distribution of all layers and heads in an LLM, we experiment with open-weight LLMs. We choose two popular instruction-tuned models: Mistral 7B (Jiang et al., 2023) and Llama-3.1 8B (Dubey et al., 2024)[2]. We note that, while we only test ICR with open-weight LLMs in this paper, ICR can also be implemented with proprietary models if their vendors choose to.

**Datasets & Metrics.** We evaluate our method on both single-hop information retrieval benchmarks, which emphasizes estimating the documents' semantic relevance, and multi-hop retrieval tasks, which demands information integration across multiple documents.

In single-hop evaluations, we experiment on TREC (Craswell et al., 2020) and nine public datasets in the BEIR benchmark (Thakur et al., 2021), including TREC-COVID (Voorhees et al., 2021), NFCorpus (Boteva et al., 2016),DBPedia-entity (Hasibi et al., 2017), SciFact (Wadden et al., 2020), SciDocs (Cohan et al., 2020), FiQA (Maia et al., 2018), FEVER (Thorne et al., 2018), Climate-FEVER (Diggelmann et al., 2020), and NQ (Kwiatkowski et al., 2019). Following Sun et al. (2023), we re-rank the top 100 documents returned by BM25(Robertson & Zaragoza, 2009) and report nDCG@10. We apply a sliding window of size 20 and stride 10 for RankGPT. For ICR, we directly re-rank all documents.

In multi-hop evaluations, we sample 1000 queries from three popular multi-hop question answering datasets as in Gutiérrez et al. (2024), including MuSiQue (answerable) (Trivedi et al., 2022),

---

[2]More specifically, we use Mistral-7B-instruct-v0.2 and Llama-3.1-8B-instruct.

Table 1: nDCG@10 on the BEIR benchmark. SR: Success rate, the chance that a re-ranker's output format is correct. We bold the best performance for each task with each base LLM.

| | DL 19 | DL 20 | TREC-COVID | NF Corpus | DBPedia-Entity | SciFact | SciDocs | FiQA | FEVER | Climate-FEVER | NQ | Micro-Avg | Macro-Avg | SR |
|---|---|---|---|---|---|---|---|---|---|---|---|---|---|---|
| BM25 | | | 59.5 | 32.2 | 31.8 | 67.9 | 14.9 | 23.6 | 65.1 | 16.5 | 30.6 | 44.7 | 40.1 | - |
| RankGPT (Mistral 7B) | 51.3 | 49.2 | 60.6 | 31.0 | **32.9** | 65.5 | 14.7 | 24.1 | 62.7 | 16.7 | 32.6 | 44.0 | 40.1 | 34.3% |
| ICR (Mistral 7B) | **59.2** | **53.6** | **63.9** | **33.2** | 31.4 | **72.4** | **16.2** | **31.0** | **79.8** | **20.6** | **51.0** | **57.3** | **46.6** | 100% |
| RankGPT (Llama-3.1 8B) | **64.9** | **63.9** | 72.6 | 33.7 | **41.4** | 68.4 | **17.9** | 31.5 | 66.7 | 22.7 | 51.1 | 52.0 | 48.6 | 99.2% |
| ICR (Llama-3.1 8B) | 55.7 | 51.9 | **72.8** | **34.7** | 35.3 | **76.1** | 17.1 | **38.1** | **84.5** | **23.1** | **51.2** | **60.4** | **49.1** | 100% |
| RankGPT (GPT-3.5 turbo) | 64.0 | 62.2 | 74.1 | 32.8 | 42.0 | 64.1 | 15.4 | 30.5 | 72.6 | 22.4 | 49.7 | 54.0 | 44.9 | 98.2% |
| RankGPT (GPT-4o mini) | 67.8 | 66.9 | 79.1 | 38.0 | 44.9 | 77.0 | 20.1 | 41.7 | 80.1 | 24.3 | 56.0 | 60.5 | 51.2 | 95.5% |

2WikiMultiHopQA (Ho et al., 2020), and HotpotQA (Yang et al., 2018). Following Gutiérrez et al. (2024), we re-rank the top 20 retrieval results returned by ColBERT v2 (Santhanam et al., 2022) and measure recall@2 and recall@5. We also report the ranking success rate for RankGPT, which measures the chance that it outputs a correctly formatted ranking.

We use the QA style prompt for multi-hop tasks, TREC-DL19, TREC-DL20, and four tasks in BEIR, including TREC-COVID, DBPedia-Entity, FiQA, and NQ. For fair comparison, we use the official prompt for RankGPT and report ColBERT v2 results if the output ranking is unusable.

## 4.2 SINGLE-HOP RE-RANKING

Compared to the proprietary GPT-3.5 Turbo and GPT-4o mini, open-weight LLMs are generally more limited in their instruction following abilities, as shown by their substantially lower performance and success rate when used with RankGPT. In contrast, ICR guarantees producing a usable ranking with any base LLM and delivers strong performance with open-weight LLMs.

Results in Table 1 demonstrates the strong performance of ICR on the single-hop re-ranking tasks, outperforming RankGPT by 13.3 (8.4) points in micro-average score and 6.5 (0.5) points in macro average when used with Mistral 7B (Llama-3.1 8B). ICR's performance advantage is more prominent on Mistral, which has noticeably weaker instruction-following capabilities compared to Llama-3.1, as shown by its significantly lower RankGPT performance and success rate. This indicates that ICR can effectively leverage re-ranking signals from LLMs even when the model is weaker at text generation. Impressively, ICR (Mistral 7B) outperforms RankGPT (GPT-3.5-turbo) by 3.3 points in micro-average and 1.7 points in macro-average. Moreover, ICR (Llama-3.1 8B) matches RankGPT (GPT-4o-mini) in micro-average (60.4 vs 60.5 points) and is slightly behind by 2.1 points in macro-average. These results highlight the effectiveness of ICR in unleashing the potential of open-weight LLMs by leveraging their internal signals. ICR substantially improves the results of BM25. We show that ICR can also work with stronger retrievers, such as ColBERTv2, in Appendix E.

We observe that ICR strongly outperforms RankGPT with both LLMs on three single-hop datasets: FiQA, SciFact and FEVER. Notably, on FEVER, RankGPT with GPT-4o mini (80.1 points) is closely matched by ICR with Mistral 7B (79.8 points), and further outperformed by ICR with Llama-3.1 8B (84.5 points). As will be detailed in Section 5.2, ICR's exceptional performance on these datasets comes from their need to leverage more complex contextualization signals in addition to similarity cues. In FIQA, the meaning of many retrieved passages can only be fully determined when combined with their respective queries, increasing the task's complexity. In SciFact and FEVER, however, relevance is determined by whether a passage supports or contradicts a specific fact, thus they also require reasoning capabilities.

Despite its high performance on most datasets, we also notice that RankGPT consistently performs better on DBPedia-Entity. We attribute much of ICR's disadvantage on DBPedia-Entity to the base LLMs' lexical bias, which hurts ICR's performance when distractor documents have high lexical overlap with the query, which is common in DBPedia-Entity (See Appendix C.1 for more details). We also surprisingly observe that ICR (llama-3.1 8B) underperforms ICR (Mistral 7B) and noticeably falls behind RankGPT (Llama 3.1 8B) on TREC DL19 and DL20. We suspect Llama-3.1 might be fine-tuned on relevant re-ranking data for MSMARCO, the data source of TREC, which leads to high re-ranking success rate but may skew its attention distribution.

## 4.3 MULTI-HOP RE-RANKING

Table 2: Re-ranking performance on the multi-hop test set. SR: Success rate, the chance that a re-ranker's output format is correct. We bold the best performance for each task and base LLM.

|  | MuSiQue | | 2Wiki | | HotpotQA | | Average | | SR |
|---|---|---|---|---|---|---|---|---|---|
|  | R@2 | R@5 | R@2 | R@5 | R@2 | R@5 | R@2 | R@5 |  |
| ColBERT v2 | 37.9 | 49.2 | 59.2 | 68.2 | 64.7 | 79.3 | 53.9 | 65.6 | - |
| RankGPT (Mistral 7B) | 36.2 | 48.7 | 56.3 | 67.1 | 64.8 | 79.8 | 52.4 | 65.2 | 20.5% |
| ICR (Mistral 7B) | **39.5** | **52.6** | **62.7** | **71.3** | **71.2** | **84.3** | **57.8** | **69.4** | 100% |
| RankGPT (Llama-3.1 8B) | 39.3 | 51.4 | 60.8 | 72.0 | 69.9 | 84.8 | 56.6 | 69.4 | 98.2% |
| ICR (Llama-3.1 8B) | **44.3** | **55.9** | **67.0** | **72.6** | **77.3** | **86.2** | **62.9** | **71.6** | 100% |
| RankGPT (GPT-3.5 Turbo) | 38.7 | 54.1 | 61.6 | 72.8 | 71.8 | 84.9 | 57.4 | 70.6 | 98.6% |
| RankGPT (GPT-4o mini) | 45.7 | 57.0 | 71.6 | 75.1 | 80.6 | 88.8 | 66.0 | 73.6 | 93.7% |

Table 3: All-Recall@5 on multi-hop test set. ICR shows substantial improvements over ColBERT v2 on all datasets. We bold the best performance for each task and base LLM.

|  | MuSiQue | 2Wiki | HotpotQA | Avg |
|---|---|---|---|---|
| ColBERT v2 | 16.1 | 37.1 | 59.0 | 33.5 |
| RankGPT (Llama3.1 8B) | 20.8 | **46.7** | 71.2 | 46.2 |
| ICR (Llama3.1 8B) | **23.5** | 44.6 | **75.7** | **47.9** |
| RankGPT (Mistral 7B) | 15.9 | 38.4 | 62.5 | 38.9 |
| ICR (Mistral 7B) | **21.3** | **43.4** | **73.5** | **46.1** |

In multi-hop evaluations, we observe similar trends as in single-hop experiments. Table 2 shows that both RankGPT and ICR substantially improve the retrieval results of ColBERT v2 on all three multi-hop test sets, with ICR outperforming RankGPT by 5.4%-6.3% in Recall@2 and 2.2%-4.2% in Recall@5 on average. Impressively, ICR (Mistral 7B) matches RankGPT (GPT-3.5 Turbo) (57.8% vs 57.4% in R@2 and 69.4% vs 70.6% in R@5). ICR (Llama-3.1 8B) further comes close to RankGPT (GPT-4o mini) (62.9% vs 66.0% in R@2 and 71.6% vs 73.6% in R@5), making it a competitive alternative to proprietary models for multi-hop re-ranking.

To better understand the performance improvement on multi-hop re-ranking, we additionally report the All-Recall@5 performance following Gutiérrez et al. (2024), which measures the rate of a retrieval model ranking all relevant documents for a query within top-5. Table 3 shows that ICR substantially improves All-Recall@5 on all three multi-hop tasks. This suggests that ICR can better understand the relationships between documents and perform multi-hop information integration. We provide a more detailed analysis in Section 5.2.

## 4.4 SCALING TREND OF SPEED AND PERFORMANCE

As previously discussed, re-ranking $N$ documents using RankGPT requires at least $O(N)$ forward passes. In contrast, ICR only requires two ($O(1)$) forward passes. We empirically compare the speed and performance of ICR and RankGPT on the Sci-Fact dataset with 300 queries, under the same settings as above. We note that the choice of dataset should not affect our conclusion. We test the scaling trend when re-ranking $K$ documents with K∈ {20, 40, 60, 80, 100}. This test is performed on a single NVIDIA RTX A6000 Ada GPU. We use Llama-3.1 8B as the base LLM and truncate the documents to 100 words per GPU memory limitations.

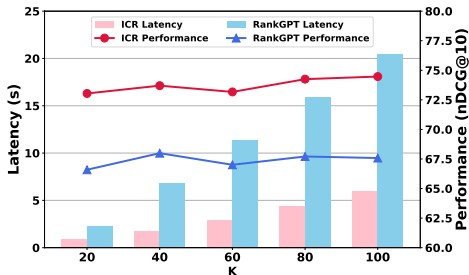

Figure 3: Scaling trend of performance and average latency on the SciFact dataset.

Table 4: Ablation studies on single-hop and multi-hop re-ranking. We report nDCG@10 on SciFact and FiQA, as well as recall@5 on MuSiQue and 2WikiMultihopQA.

| | Single-hop | | | Multi-hop | | |
| | SciFact | FiQA | Macro-Avg | MuSiQue | 2Wiki | Macro-Avg |
|---|---|---|---|---|---|---|
| ICR (Llama-3.1 8B) | **76.1** | **38.1** | **57.1** | **55.9** | **72.6** | **64.3** |
| – Calibration | 71.3 | 28.8 | 50.0 | 52.1 | 64.4 | 58.2 |
| – Aggregation | 68.2 | 24.4 | 46.3 | 41.9 | 52.6 | 47.2 |
| – Both | 60.9 | 20.1 | 40.5 | 41.1 | 50.3 | 45.7 |

We implement ICR with the Transformers library (Wolf et al., 2020) by extracting attention weights from the `forward()` function. We implement RankGPT using vLLM (Kwon et al., 2023), a popular LLM inference infrastructure that is much faster than Transformers to ensure a practical comparison.[3] Figure 3 shows that ICR is more than two times faster than RankGPT with a lower rate of growth when K becomes larger, despite having a less efficient implementation. Meanwhile, we notice ICR's performance keeps improving as K grows, while RankGPT saturates at $K = 40$.

## 4.5 ABLATION STUDY

In this section, we evaluate the effectiveness of attention aggregation and score calibration in ICR. We conduct ablation studies on SciFact, FiQA, MuSiQue, and 2WikiMultiHopQA. When both components are removed, ICR ranks the documents only based on the attention weights of the last query token, i.e., equivalent to attention sorting (Peysakhovich & Lerer, 2023).

As shown in Table 4, both attention aggregation and score calibration are crucial for ICR's high performance. The huge gain brought by attention aggregation (13.7 points in single-hop and 17.1 percent in multi-hop on average) shows the importance of considering all query tokens rather than only the last token. The calibration process also has a big performance impact (9.3 points in single-hop and 6.1 percent in multi-hop on average). We analyze why calibration helps in Section 5.1.

## 5 ANALYSIS & DISCUSSION

In this section, we first analyze why score calibration is effective (Section 5.1). Then, we investigate how ICR leverages more complex signals in LLMs for complex retrieval tasks using Llama-3.1 8B (Section 5.2). We analyze ICR's behavior based on token-level ranking scores of documents.

### 5.1 WHY DOES CALIBRATION IMPROVE IN-CONTEXT RE-RANKING?

To understand the effectiveness of the proposed calibration method, we analyze the distribution of calibration score $S_{cal}$ on multi-hop datasets. As the calibration query is irrelevant to any document, an unbiased re-ranker should assign equal ranking scores for all documents. Therefore, calibration scores that deviate from a uniform distribution can reflect the base LLM's intrinsic biases.

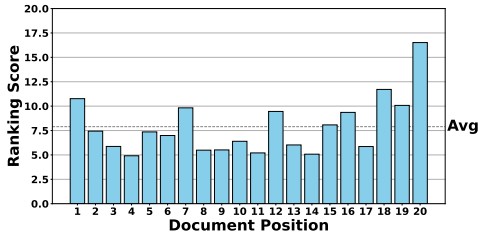

Figure 4: Average calibration scores of ICR at different document input positions.

Table 5: An example of token-level calibration score in ICR (Llama-3.1 8B). Deeper color indicates higher calibration score.

Indo -P ak istani wars and conflicts C
The Kashmir issue has been the main cause , whether direct or indirect , of all major conflicts between the two countries with the exception of the Indo - Pakistani War of 197 1 where conflict originated due to turmoil in erst while East Pakistan ( now Bangladesh ).

[3] We note that ICR is compatible with LLM inference infrastructures like vLLM as well if an interface for getting attention weights is exposed (not available for vLLM as of now).

Table 6: In-context re-ranking scores for all tokens in a FIQA example with query "How to read bond yield quotes? What do the time, coupon, price, yield, and time mean?" B, R, I denote ranking from BM25, RankGPT, and ICR respectively.

| Passage | B | R | I | Support |
|---|---|---|---|---|
| The 1 month and 1 year columns show the percentage change over that period. Coupon (coupon rate) is the amount of interest paid on the bond each period. Price is the normalised price of the bond ... Yield is the interest rate that you would receive by buying at that price. ... | 6 | 10 | 1 | True |
| The 10yr bond pays coupons semi annually. The yield % is what you would get annually if you hold the bond to full term. The coupon payment won't be exactly 1.65%/2 because of how bond pricing and yields work. The yield commonly quoted does not tell you how much each interest payment is. ... | 2 | 1 | 2 | False |

We observe a strong position bias towards documents placed at the beginning and the end of the input on both Mistral and Llama-3.1, as shown in Figure 4, echoing the lost-in-the-middle issue discussed in recent literature (Liu et al., 2024). With the proposed calibration method, ICR shows improved robustness against different document input orders compared to RankGPT, which partly explains its superior performance (Quantitative results are available in Appendix D.).

Since ICR determines a document's relevance by summing ranking scores over all its tokens, token-level scores could directly reveal ICR's inner workings. In addition to the position bias, the example in Table 5 shows that Llama3.1's attention distribution is biased towards titles, entities and punctuation (Similar results with Mistral can be found in Appendix G.1). By capturing these biased signals in the LLM, calibration helps to mitigate intrinsic biases and improve re-ranking performance.

## 5.2 WHAT COMPLEX RE-RANKING SIGNALS DOES ICR LEVERAGE?

Our experiment results show that ICR dramatically outperforms RankGPT on several single-hop datasets (FiQA, FEVER, SciFact) and all multi-hop re-ranking tasks. In this section, we explain how re-ranking on these datasets requires more complex re-ranking signals than more conventional retrieval datasets and provide a comprehensive investigation of ICR's improved ability to use such signals. We use token-level ranking scores in our analysis similar to the previous section.

**Query-Passage Contextualization.** As opposed to other QA datasets in BEIR like TREC-COVID, NFCorpus and NQ, which use stand-alone text such as scientific abstracts and Wikipedia articles as their retrieval corpora, FiQA's retrieval corpus consists of human-written responses to the search queries in the dataset. As shown in the example in Table 6, this makes many passages only intelligible when their corresponding question is present, which makes re-ranking these passages a specially complex challenge that requires deeper contextualization between passages and the query.

Table 6 clearly illustrates ICR's improved detection of such query-passage contextualization. In the passage ranked $1^{st}$ by ICR, which RankGPT ranks at the $10^{th}$, the strongest signals come from the phrases "coupon (coupon rate) is", "price is", "yield is" and "time is", indicating that the model is not only detecting lexical overlap but also attending to the word "is", which is essential to answering the query. Interestingly, the distractor passage, which ICR ranks $2^{nd}$ and RankGPT ranks $1^{st}$, shows weaker and more diffused attention patterns, indicating the model's uncertainty concerning this document's relevance to the query. We provide more such examples in Appendix G.2.

**Contradiction-Based Reasoning.** ICR shows exceptionally strong performance on FEVER and SciFact, two fact verification datasets. By breaking down the performance across claim types, we find that ICR does particularly well on claims which are contradicted by the retrieved passages. On FEVER, ICR using Llama-3.1 improves performance over RankGPT on *Contradiction* examples by 26 points while only by 9 points on *Support* examples (SciFact sees a similar gap of 15 points to 5 points). We note that the evidence required for refuting claims usually have lower lexical overlap with the query and instead rely on deeper reasoning processes which must consider both the search query and the passages, resulting in a more challenging task.

In Table 7, we see that ICR can more effectively leverage these reasoning processes than RankGPT and thereby more accurately rank refuting passages. The first example in Table 7 shows a FEVER

Table 7: In-context re-ranking scores for all tokens in a FEVER example with the query "Night of the Living Dead was originally directed by Krzysztof Kieslowski." B, R, I denote ranking from BM25, RankGPT, and ICR respectively.

| Passage | B | R | I | Support |
|---|---|---|---|---|
| Night of the Living Dead is a 1968 American independent horror film , directed by George A. Romero , starring Duane Jones and Judith O'Dea . It was completed on a $ 114,000 budget and premiered October 1 , 1968 . ... | 11 | 28 | 1 | True |
| The Scar ( Blizna ) is a 1976 Polish film written and directed by Krzysztof Kieślowski and starring Franciszek Pieczka . Filmed on location in Olechów , Poland , the film is about a man put in charge of the construction ... | 1 | 1 | 4 | False |

Table 8: In-context re-ranking scores for all tokens in a MuSiQue example with the query "Who was the spouse of a leading speaker against slavery and publisher of an antislavery newspaper?" The bridge entity needed to reach all supporting documents is "Frederick Douglass."

| First Hop | Second Hop |
|---|---|
| The North Star was a nineteenth - century anti-slavery newspaper published from the Talman Building in Rochester, New York by abolitionist Frederick Douglass. | Helen Pitts Douglass (1838–1903) was an American suffragist and abolitionist, known for being the second wife of Frederick Douglass. She also created the Frederick Douglass Memorial and Historical Association. |

example in which ICR ranks the passage "Night of the Living Dead" as the most relevant one by attending to the true director "George A. Romero", since it refutes the claim that it was directed by "Krzysztof Kieslowski". On the other hand, both RankGPT and BM25 rank another passage mentioning a film directed by "Krzysztof Kieslowski" at the top, probably due to its high lexical similarity. We provide more such examples for both FEVER and SciFact in Appendix G.3.

**Multi-Hop Information Integration.** ICR obtains strong performance on multi-hop retrieval tasks, which requires integrating information across multiple passages. Through token-level analysis, we show that ICR assigns high ranking scores to *bridge entity* tokens that are essential in the integration process. We demonstrate this phenomenon in Table 8 by showing how ICR (Llama-3.1 8B) scores each token on an example from MuSiQue (See more examples in Appendix G.4.). The bridge entity "Frederick Douglass" provides a stronger than average signal in both the first hop and second hop documents, suggesting that ICR is leveraging the LLM's multi-hop reasoning signals.

## 6 CONCLUSION

We propose in-context re-ranking (ICR), an efficient re-ranking method based on the attention distributions of LLMs. Compared to generative methods, it only requires $O(1)$ forward passes for re-ranking $N$ documents, instead of at least $O(N)$ forward passes, significantly cutting the latency of re-ranking. Through comprehensive experiments, we demonstrate the strong zero-shot performance of ICR on both standard single-hop information retrieval benchmarks and challenging multi-hop re-ranking tasks. Compared to RankGPT, ICR is particularly strong at handling more complex retrieval tasks. By directly leveraging internal signals within LLMs, ICR substantially boosts the re-ranking performance of open-weight LLMs to match strong proprietary models.

Our analysis further reveals the abundant re-ranking signals carried by attention weights, and that ICR can effectively leverages such signals especially in challenging tasks. The evidence presented in our detailed analysis concerning ICR's improved ability to detect complex re-ranking signals strongly supports our hypothesis that the abundant re-ranking signals that exist within LLMs might not be optimally leveraged through generative approaches. These findings also highlight the potential of developing better ways to leverage open-weight models.

ACKNOWLEDGEMENTS

The authors would like to thank Yao Fu and Kai Zhang for their constructive feedback. We also appreciate the thoughtful comments from colleagues of the OSU NLP group. This work is supported in part by NSF OAC 2112606 and NIH R01LM014199. The views and conclusions contained herein are those of the authors and should not be interpreted as representing the official policies, either expressed or implied, of the U.S. government. The U.S. government is authorized to reproduce and distribute reprints for government purposes notwithstanding any copyright notice herein.

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

APPENDICES

## A  COMPLEXITY OF LLM-BASED RE-RANKING METHODS

For a fair comparison of complexity in terms of forward passes (FPs), we assume the LLM used for re-ranking has a long enough context window to process all input documents at once. We consider two types of forward passes in the LLM inference process: those in the prefill phase and those in the decode phase. The complexity of a re-ranking method is the sum of the number of forward passes needed in prefill and decode phases.

We note that, the generation targets of generative methods, such as document identifiers or ranking scores, are typically much shorter than the input context. Therefore, we can approximately treat the cost of FPs in the decode phase as constant.

| Method | #API calls | #Prefill FPs per API call | #Decode FPs per API call | Total FPs |
|---|---|---|---|---|
| pointwise | $O(N)$ | $O(1)$ | $O(1)$ | $O(N)$ |
| pairwise | $O(N) \sim O(N^2)$ | $O(1)$ | $O(1)$ | $O(N) \sim O(N^2)$ |
| listwise | $O(1)$ | $O(1)$ | $O(N)$ | $O(N)$ |
| ICR (ours) | $O(1)$ | $O(1)$ | N/A | $O(1)$ |

Table A.1: Complexity of different re-ranking methods in terms of forward passes. $N$ is the number of documents to re-rank.

We list the number of API calls and FPs in each phase per API call required by different re-ranking methods in Table A.1. As illustrated, pointwise and pairwise methods require at least $O(N)$ LLM API calls, leading to a total of $O(N)$ FPs in both prefill and decode phases. Listwise re-ranking methods only invoke the LLM API once, but needs to use $O(N)$ FPs in the decode phase for generating a list of $N$ document identifiers. Similar to listwise re-ranking, ICR also only needs $O(1)$ LLM API calls. However, by fully leveraging the signals produced in the prefill stage, ICR avoids the decode stage, leading to an overall complexity of only $O(1)$ FPs.

## B  PROMPTS

We present exmaple prompts used in ICR and RankGPT in our experiments.

### B.1  ICR PROMPT

Table B.2 shows an example prompt for ICR on HotpotQA using the QA style instruction. Table B.3 shows an example prompt for ICR on FEVER using the IE style instruction. The prompt is wrapped by LLM-specific special tokens `<prefix>` and `<suffix>` to correctly leverage their instruction following abilities, such as `[INST]` and `[/INST]` for Mistral 7B.

### B.2  RANKGPT PROMPT

Table B.4 shows an example prompt for RankGPT on TREC-COVID. The prompt is also wrapped by LLM-specific special tokens. In addition, we add `Ranked Passages:  [` after `<suffix>` to increase the ranking success rate. Still, open-weight LLMs often fails to generate a usable ranking.

Table B.2: An example prompt for ICR on HotpotQA using the QA style instruction.

```
<prefix>Here are some paragraphs.  Please answer the question based on the relevant
information in the paragraphs.

[1] Document 1 Chris Sale
Christopher Allen Sale (born March 30, 1989), nicknamed The Condor, is an American
professional baseball pitcher for the Boston Red Sox of Major League Baseball
(MLB). Sale was selected 13th overall in the 2010 Major League Baseball draft by
the Chicago White Sox and made his MLB debut with them in 2010.  He is a six-time
MLB All-Star, and he led the American League in strikeouts in 2015.  Prior to playing
professionally, he played college baseball for Florida Gulf Coast University.

...

[20] Klay Thompson Klay Alexander Thompson (born February 8, 1990) is an American
professional basketball player for the Golden State Warriors of the National
Basketball Association (NBA). The son of former NBA player Mychal Thompson, he
played college basketball for three seasons at Washington State University, where
he was a two-time first-team all-conference selection in the Pac-10.  Thompson was
selected in the first round of the 2011 NBA draft by Golden State with the 11th
overall pick.  In 2014, he and teammate Stephen Curry set a then NBA record with 484
combined three-pointers in a season, as the pair were given the nickname the "Splash
Brothers".  Thompson is a three-time NBA All-Star and a two-time All-NBA Third Team
honoree.  In 2015, he helped lead the Warriors to their first NBA Championship since
1975.  Thompson helped the Warriors return to the NBA Finals for a third straight
year in 2017, winning his second NBA Championship.

Query:  What relationship does Fred Gehrke have to the 23rd overall pick in the 2010
Major League Baseball Draft?<suffix>
```

Table B.3: An example prompt ICR on FEVER using the IE style instruction.

```
<prefix>Here are some paragraphs.  Please find information that are relevant to the
query.

[1] Ukraine and the United Nations
The Ukraine was one of the founding members of the United Nations when it joined
in 1945 as the Ukrainian Soviet Socialist Republic along with the Byelorussian
Soviet Socialist Republic signed the United Nations Charter when they were part
of the Soviet Union .  After the dissolution of the Soviet Union in 1991 , the
newly-independent Ukraine retained its seat

...

[20] Council of People's Ministers
The Council of People 's Ministers of Ukraine was the main executive institution
of the Ukrainian People 's Republic .  Its duties and functions were outlined in
the Chapter V of the Constitution of the Ukrainian National Republic .  It was
reorganized out of the General Secretariat of Ukraine upon the proclamation of the
4th Universal and Independence on January 25 , 1918 after the return of the Ukrainian
delegation from the preliminary peace talks from Brest-Litovsk .  At the preliminary
talks in Brest , Ukraine was recognized as an equal-rightful participant and was
scheduled to finalized the treaty on February 9 , 1918 .  Until the end of the month
January 1918 the member of the former General Secretariat continued to serve as full
pledged ministers .

Query:  Ukrainian Soviet Socialist Republic was a founding participant of the
UN.<suffix>
```

## C  LIMITATIONS OF ICR

### C.1  ICR STILL SUFFERS FROM LEXICAL BIAS

Given that ICR sometimes underperforms RankGPT in our experiments, we use the same analysis tools for understanding its limitations.

Firstly, we find that ICR signal is highly concentrated: most of the re-ranking score comes from a small number of tokens in a few documents. This characteristic leads to most documents receive too little scores to be differentiated. Secondly, we find that this signal is mostly concentrated in phrases that are lexically similar to the query.

Table B.4: An example prompt for RankGPT on TREC-COVID.

```
<prefix>This is an intelligent assistant that can rank passages based on their
relevancy to the query.

The following are 20 passages, each indicated by number identifier [].  I can rank
them based on their relevance to query:  "what is the origin of COVID-19"

[1]Origin of Novel Coronavirus (COVID-19):  A Computational Biology Study using
Artificial Intelligence

Origin of the COVID-19 virus has been intensely debated in the scientific community
since the first infected cases were detected in December 2019.  The disease has
caused a global pandemic, leading to deaths of thousands of people across the world
and thus finding origin of this novel coronavirus is important in responding and
controlling the pandemic...

...

[20] Papa Giovanni XXIII Bergamo Hospital at the time of the COVID-19 outbreak:
letter from the warfront.

In early December 2019, the 2019 novel coronavirus (COVID-19) was identified as
the agent responsible for the first pneumonia cases of unknown origin in Wuhan,
the capital of the Hubei region in China.  The virus has been identified as a novel
enveloped RNA betacoronavirus2 , that has been promptly named SARS-CoV-2 (severe
acute respiratory syndrome coronavirus 2).  The World Health Organization (WHO), on
January 12, 2020 declared the COVID-19 a public health emergency of international
concern.  On March 11, the WHO made the assessment that COVID-19 can be characterized
as a pandemic.

The search query is:  "what is the origin of COVID-19".  I will rank the 20 passages
above based on their relevance to the search query.  The passages will be listed in
descending order using identifiers, the most relevant passages should be listed first
and the output format should be [] > [] > etc, e.g., [1] > [2] > etc.  Be sure to
list all 20 ranked passages and do not explain your ranking until after the list is
done.<suffix>Ranked Passages:  [
```

Table C.1: In-context re-ranking scores for all tokens in several DBPedia-Entity examples. B, R, and I denote ranking from BM25, RankGPT, and ICR respectively.

| Query | Passage | B | R | I | Support |
|---|---|---|---|---|---|
| Give me all launch pads operated by NASA. | [ 71 ] Launch pad Ċ A launch pad is an above -ground platform from which a rocket -powered missile or space vehicle is vertically launched . A space port ( or launch complex ) is a facility which includes , and provides required support for , one or more launch pads . | 29 | 31 | 1 | False |
| | [ 92 ] Cape Can av eral Air Force Station Launch Complex 19 Ċ Launch Complex 19 ( LC - 19 ) is a deactivated launch site on Cape Can av eral Air Force Station , Florida used by NASA to launch all of the Gemini manned space fl ights . It was also used by unmanned Titan I and Titan II missiles .L C - 19 was in use from 195 9 to 196 6 , during which time it saw 27 launches , 10 of which were manned . The first use of LC - 19 was on August 14 , 195 9 . This was a Titan I and the mission was declared a failure after the rocket exploded while still on the pad . | 8 | 2 | 16 | True |

This bias leads to ICR's lower performance in standard information retrieval tasks where document relevance depends on much more than pure lexical similarity. Additionally, ICR's performance on DBPedia-Entity and 2WikiMultihopQA suffers from this limitation due to its entity-centric query construction that leads to many distractor documents containing entities with high lexical overlap with the query, and therefore getting higher scores from ICR.

While this lexical bias appears to overpower other signals in ICR, it remains unclear whether more comprehensive document understanding signal will be unveiled if we adjust the aggregation strategy. We hope to explore this interesting direction in future work.

## C.2 OTHER LIMITATIONS

This work is limited to using decoder-only language models, due to their popularity and strong performance. We leave extending ICR to other language model architectures to future work.

The proposed method relies on the availability of attention weights from the base LLM, thus limiting its applicability to proprietary LLMs with only API access. While in-context re-ranking shows promising performance on recently developed open-weight LLMs, RankGPT with proprietary LLMs still has the best performance. It would be interesting to test the performance of ICR on stronger models with state-of-the-art performance.

Our mechanistic analysis of ICR reveals its vulnerability to syntactic similarities. This poses a potential risk in applying ICR to practical usage. For example, content providers could inject certain phrases to trick ICR for better ranking. Future work could further enhance ICR's robustness.

## D ROBUSTNESS TO INPUT DOCUMENT ORDER

Table D.2: Re-ranking performance given different document input orders. We report nDCG@10 on Trec-Covid and Recall@5 on MuSiQue. For random order, we report average performance and standard deviation across 5 different random seeds.

|  | Order | Covid | MuSiQue |
|---|---|---|---|
| Retriever | - | 59.5 | 37.9 |
| RankGPT (Llama3.1 8B) | Retriever | 72.6 | 52.6 |
| ICR (Llama3.1 8B) | Retriever | 66.8 | 56.8 |
| RankGPT (Llama3.1 8B) | Reversed Retriever | 72.9 | 50.5 |
| ICR (Llama3.1 8B) | Reversed Retriever | 72.8 | 55.9 |
| RankGPT (Llama3.1 8B) | Random | 65.6 (2.58) | 49.8 (0.72) |
| ICR (Llama3.1 8B) | Random | **69.8 (1.66)** | **55.6 (0.45)** |

To study the robustness of different methods against position bias, we present the re-ranking performance of RankGPT and ICR on single-hop and multi-hop datasets with different input orders. We evaluate re-ranking performance when input documents are in the order returned by the retriever, in reversed order of the retriever, or in random order. In the random order setting, we use 5 random seeds and report average performance as well as standard deviation.

As shown in Table D.2, both RankGPT and ICR work the best when the input is ranked by the retriever. We observe that ICR works better when using the reverse order on single-hop datasets, possibly due to the recency bias of LLMs. When the input is randomly shuffled, ICR shows considerably higher performance and lower standard deviation than RankGPT, indicating stronger robustness.

## E ICR + COLBERTV2 RESULTS ON BEIR

Our main experiments showed the effectiveness of ICR when re-ranking results of BM25. Here, we demonstrate that ICR can also improve results produced by stronger dense retrieval methods, such as ColBERT v2. We report results with Llama-3.1 8B on the BEIR benchmark in Table E.3.

Table E.3: Llama-3.1 8B's re-ranking performance (nDCG@10) on the BEIR benchmark with different retrievers. SR: Success rate, the chance that a re-ranker's output format is correct. We bold the best performance for each task with each retriever.

|  | TREC-COVID | NF Corpus | DBPedia-Entity | SciFact | SciDocs | FiQA | FEVER | Climate-FEVER | NQ | Micro-Avg | Macro-Avg | SR |
|---|---|---|---|---|---|---|---|---|---|---|---|---|
| BM25 | 59.5 | 32.2 | 31.8 | 67.9 | 14.9 | 23.6 | 65.1 | 16.5 | 30.6 | 44.6 | 38.0 | - |
| + RankGPT | 72.6 | 33.7 | **41.4** | 68.4 | **17.9** | 31.5 | 66.7 | 22.7 | 51.1 | 51.9 | 45.1 | 99.2% |
| + ICR | **72.8** | **34.7** | 35.3 | **76.1** | 17.1 | **38.1** | **84.5** | **23.1** | **51.2** | **60.5** | **48.1** | 100% |
| ColBERT v2 | 65.7 | 33.2 | 44.3 | 67.5 | 15.1 | 35.6 | 78.3 | 18.3 | 54.5 | 57.6 | 45.8 | - |
| + RankGPT | **77.4** | 34.1 | **47.6** | 66.1 | 17.4 | 35.6 | 67.7 | 21.9 | **60.1** | 54.7 | 47.5 | 99.2% |
| + ICR | 74.6 | **35.8** | 40.5 | **74.4** | 17.4 | **40.9** | **86.6** | **22.7** | 59.7 | **63.7** | **50.3** | 100% |

When used with ColBERTv2, ICR still brings a substantial performance improvement of 6.1 points in micro-average and 4.5 points in macro-average. We also notice that ICR shows the largest advantage over RankGPT in more reasoning-intensive tasks, such as SciFact, FiQA, and FEVER, while underperforms on DBPedia-Entity. This is consistent with our observations based on BM25.

## F    COMPARISON WITH SUPERVISED AND SETWISE APPROACHES

To understand how ICR compares with other more costly re-ranking methods, such as the supervised fine-tuned RankVicuna (Pradeep et al., 2023a) and the more complex setwise re-ranking method (Zhuang et al., 2024), we compare ICR's performance with them in Table F.4.

Table F.4: Re-ranking performance (nDCG@10) on four BEIR datasets. *: Results are copied from (Zhuang et al., 2024)

|  | NFCorpus | DBPedia-Entity | SciFact | Trec-Covid |
|---|---|---|---|---|
| BM25 | 32.2 | 31.8 | 67.9 | 59.5 |
| RankGPT (Llama 3.1 8B) | 33.7 | 41.4 | 68.4 | 72.6 |
| ICR (Llama 3.1 8B) | 34.7 | 35.3 | **76.1** | 72.8 |
| RankVicuna | 34.7 | **44.4** | 71.3 | **80.7** |
| Setwise.heapsort (T5-XL)* | 35.2 | 42.8 | 67.7 | 75.7 |
| Setwise.bubblesort (T5-XL)* | **35.3** | 43.8 | 69.1 | 75.65 |
| RankGPT (3.5-turbo) | 32.8 | 42.0 | 64.1 | 74.1 |
| RankGPT (4o-mini) | 38.0 | 44.9 | 77.0 | 79.1 |

We notice that RankVicuna and setwise methods show better performance on Trec-Covid, NFCorpus and DBPedia, as expected. Both methods incur additional cost in training (RankVicuna) or inference time (Setwise methods). Similar to our findings in the main text, ICR shows impressive performance on SciFact, outperforming supervised methods without specialized training.

## G    MORE QUALITATIVE EXAMPLES

### G.1    INTRINSIC BIASES OF MISTRAL 7B

We present the intrinsic biases captured by the calibration process with Mistral 7B in the same way as in Section 5.1. Similar to Llama-3.1 8B, Mistral 7B also exhibit a strong position bias to both sides of the context window, shifted to the front. Mistral also demonstrate a strong bias towards document identifiers, title, entities, and punctuation.

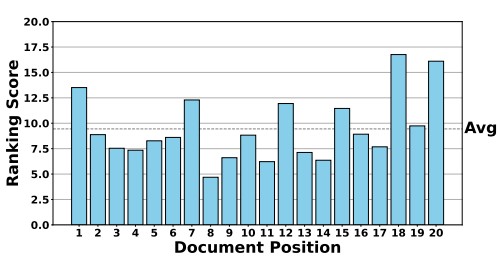

Figure G.1: Calibration Score of Mistral 7B.

Table G.5: An example of token-level calibration score in ICR (Mistral 7B). Deeper color indicates higher score.

[ 2 0 ] Long Sp ru ce Gener ating Station ⟨0x0A⟩ It was Man it oba H ydro ' s fourth generating station to be built on the Nelson River , which flows from Lake W inn ipe g to Hudson Bay . The station was built on Long Sp ru ce Rap ids . The site is approximately east of Gill am , Man it oba and is down stream of Man it oba H ydro ' s K ett le Gener ating Station .

### G.2    QUERY-PASSAGE CONTEXTUALIZATION

As discussed in Section 5.2, due to FIQA's construction, many of its passages are not easily interpreted without context from their corresponding query, making the re-ranking process more challenging. In Table G.6, we illustrate FIQA's more complex nature with three additional examples that also include ICR and RankGPT performance as well as ICR's per-token re-ranking scores. In the first query, we see that ICR detects that Llama-3.1 strongly attends to the question's answer: "Syrian refugee situation in Europe", even though the phrase has no direct semantic relation to the query. In contrast, RankGPT ranks this document at position 22 and elevates a distractor to the top position even though it is not related to the query since it mentions the Berkshire Hathaway shareholders

meeting directly. Similarly, in the second and third queries, ICR detects that Llama-3.1 attends to tokens and phrases which are directly related to answering the query: stating that nothing would happen to the user's bank account in the first query and that providing services to Germany from Poland (another EU country) without paying taxes would be allowed. On the other hand, RankGPT is unable to re-rank these passages correctly and instead opts for passages which have significant semantic overlap with the query but are actually answering completely different queries.

Table G.6: In-context re-ranking scores for all tokens in several FIQA examples. B, R, and I denote ranking from BM25, RankGPT, and ICR respectively.

| Query | Passage | B | R | I | Support |
|---|---|---|---|---|---|
| What to ask Warren Buffet at the Berkshire Hathaway shareholder meeting? | For whatever it's worth, when I went to the meeting a couple of years ago, the question and answer segment is mostly students asking how to pick a stock or what book they should read. I'm sure someone else will ask but it would be interesting to hear their take on the Syrian refugee situation in Europe and how it may impact the EU in general. ... | 61 | 22 | 1 | True |
| | "'Autonomous vehicles would hurt us if they spread to trucks"" Buffett told shareholders at Berkshire Hathaway's (BRK.A) annual meeting in May. If self-driving trucks become predominant on the roads, it could steal business from Berkshire owned railroad Burlington Northern, Buffett hinted. >Buffett acknowledged that autonomous cars are ""coming"", and could also ""hurt"" Berkshire's insurance business Geico. | 1 | 1 | 6 | False |
| If the U.S. defaults on its debt, what will happen to my bank money? | You must mean the current debt ceiling debacle. The meaning of it is: US government is constantly borrowing money (by issuing treasury bonds) and constantly repaying some of the bonds that come to maturity, ... Going back to your bank account, most probably nothing would happen to the money you store there. Even if the bank had invested 100% of the money in US treasury bonds (which doesn't really happen) ... | 16 | 20 | 1 | True |
| | Andrew Lilico has a likely scenario for when Greece defaults on its sovereign debt: What happens when Greece defaults. Here are a few things: Every bank in Greece will instantly go insolvent. The Greek government will nationalise every bank in Greece. The Greek government will forbid withdrawals from Greek banks. ... | 1 | 1 | 6 | False |
| Does freedom to provide services allow me contracting in Germany without paying taxes there (but in my home EU country)? | You're free to provide services, but if you stay in one country for more than half a year - you're generally considered to be its resident for tax purposes. Germany is no exception to the rule, in fact - this is true to almost any country in the world. If you provide the services from Poland, and never set foot in Germany - they won't say a word. | 16 | 4 | 1 | True |
| | "You'll need to read carefully the German laws on tax residency, in many European (and other) tax laws the loss of residency due to absence is conditioned on acquiring residency elsewhere. But in general, it is possible to use treaties and statuses so that you end up not being resident anywhere, but it doesn't mean that the income is no longer taxed. ... Keep in mind, that the treaty has ""who is or was immediately before visiting a Contracting State a resident of the other Contracting State"" language in some relevant cases, so you may still apply it in the US even if no longer resident in Germany." ... | 68 | 1 | 13 | False |

## G.3 CONTRADICTION-BASED REASONING

In this section, we provide more examples of ICR's improved performance in contradiction-based reasoning examples within fact verification datasets such as FEVER and SciFact, as explained in Section 5.2. Some FEVER examples are provided in Table G.7. The first example shows that ICR leverages Llama-3.1's attention patterns on Jiang Wen's many talents as an actor, screenwriter and director to refute the the claim that he is exclusively a producer. On the other hand, RankGPT incorrectly re-ranks this document to position 13, opting instead for documents where "Jiang Wen" is mentioned in conjunction with producers because of semantic overlap. In the second example, RankGPT is strongly biased towards matching the year "1995" and thus re-ranks a completely unrelated passage to the first position. In contrast, ICR is able to detect more subtle attention patterns placed by Llama-3.1 on the year "1975", which is the true year in which Balibo started.

In Table G.8, we include some contradiction-based examples from the SciFact dataset. Although they are much more domain-specific and challenging, we find that ICR also demonstrates the enhanced ability to leverage deeper reasoning signals which we see in the FEVER dataset. In the first query, we are looking for statements related to the enzyme aPKCz and how it affects tumor growth, like the last sentence in the first passage. This sentence claims that "PKC deficiency", not PKC, causes cancer cell enhancement through the use of glutamine, effectively refuting the claim. From the attention patterns shown in the table, it seems that ICR is able to detect that the Llama-3.1 model attends to this contradictory sentence and the passage as a whole more strongly in its attempt to verify this claim, bringing it from position 8 to position 1 while RankGPT brings it down to position 9. In contrast, RankGPT elevates the passage in the second row to the first spot, most likely due to more lexical cues since the passage discusses how the KRAS protein affects glutamine metabolism and yields tumor enhancements.

The second example in this table shows a claim concerning how interferon-induced genes reduce granule cell survival when exposed to West Nile virus. Even though both the gold document and the distractor are quite similar to the claim in this case, RankGPT is unable to detect that the first passage actually finds that "interferon-stimulated genes" **increase** granule cell survival, exactly the opposite of our claim. From the attention patterns, we see that Llama-3.1 is strongly attending to this passage as a whole as well as concentrating attention on the last sentence which is the most relevant to the claim. On the other hand, we see that the attention patterns of Llama 3.1 in the distractor passage are more concentrated on lexical similarities such as the "West Nile virus".

Table G.7: In-context re-ranking scores for all tokens in several FEVER contradiction-based examples. B, R, and I denote ranking from BM25, RankGPT, and ICR respectively.

| Query | Passage | B | R | I | Support |
|---|---|---|---|---|---|
| Jiang Wen is exclusively a producer. | Jiang Wen ( born 5 January 1963 ) is a Chinese film actor , screenwriter , and director . As a director , he is sometimes grouped with the " Sixth Generation " that emerged in the 1990s . ... | 2 | 13 | 1 | True |
| | Emperor Motion Pictures ( known as EMP ) is a film producer and distributor , part of the Emperor Group . Following the 2003 box-office hits The Twins Effect and The Medallion , EMP has produced ... The Sun Also Rises and Forever Enthralled , two works by renowned Chinese auteurs Jiang Wen and Chen Kaige. ... | 32 | 1 | 2 | False |
| Balibo (film) starts in the year 1995. | Balibo (film) is a 2009 Australian war film that follows the story of the Balibo Five , a group of journalists who were captured and killed while reporting on activities just prior to the Indonesian invasion of East Timor of 1975 . ... | 1 | 19 | 1 | True |
| | Bjørnar Teigen (born 29 June 1971) is a Norwegian actor , theater director and playwright. Teigen studied acting in the Oslo National Academy of the Arts and started working as an actor in Molde 's Teatret Vårt after graduation in 1995. ... | 26 | 1 | 62 | False |

Table G.8: In-context re-ranking scores for all tokens in several SciFact contradiction-based examples. B, R, and I denote ranking from BM25, RankGPT, and ICR respectively.

| Query | Passage | B | R | I | Support |
|---|---|---|---|---|---|
| aPKCz causes tumour enhancement by affecting glutamine metabolism. | Control of Nutrient Stress-Induced Metabolic Reprogramming by PKCζ in Tumorigenesis Tumor: cells have high-energetic and anabolic needs and are known to adapt their metabolism to be able to survive and keep proliferating under conditions of nutrient stress. We show that PKCζ deficiency promotes the plasticity necessary for cancer cells to reprogram their metabolism to utilize glutamine through the serine biosynthetic pathway in the absence of glucose. [...] | 8 | 9 | 1 | True |
| | Glutamine supports pancreatic cancer growth through a Kras-regulated metabolic pathway: Cancer cells have metabolic dependencies that distinguish them from their normal counterparts. Among these dependencies is an increased use of the amino acid glutamine to fuel anabolic processes. Indeed, the spectrum of glutamine-dependent tumours and the mechanisms whereby glutamine supports cancer metabolism remain areas of active investigation. [...] Moreover, knockdown of any component enzyme in this series of reactions also results in a pronounced suppression of PDAC growth in vitro and in vivo. Furthermore, we establish that the reprogramming of glutamine metabolism is mediated by oncogenic KRAS, the signature genetic alteration in PDAC, through the transcriptional upregulation and repression of key metabolic enzymes in this pathway. [...] | 1 | 1 | 3 | False |
| Rapid up-regulation and higher basal expression of interferon-induced genes reduce survival of granule cell neurons that are infected by West Nile virus. | Differential innate immune response programs in neuronal subtypes determine susceptibility to infection in the brain by positive stranded RNA viruses: Although susceptibility of neurons in the brain to microbial infection is a major determinant of clinical outcome, little is known about the molecular factors governing this vulnerability. Here we show that two types of neurons from distinct brain regions showed differential permissivity to replication of several positive-stranded RNA viruses. Granule cell neurons of the cerebellum and cortical neurons from the cerebral cortex have unique innate immune programs that confer differential susceptibility to viral infection ex vivo and in vivo. By transducing cortical neurons with genes that were expressed more highly in granule cell neurons, we identified three interferon-stimulated genes (ISGs; Ifi27, Irg1 and Rsad2 (also known as Viperin)) that mediated the antiviral effects against different neurotropic viruses. Moreover, we found that the epigenetic state and microRNA (miRNA)-mediated regulation of ISGs correlates with enhanced antiviral response in granule cell neurons. [...] | 3 | 3 | 1 | True |
| | Beta interferon controls West Nile virus infection and pathogenesis in mice. Studies with mice lacking the common plasma membrane receptor for type I interferon (IFN-$\alpha\beta$R(-)(/)(-)) have revealed that IFN signaling restricts tropism, dissemination, and lethality after infection with West Nile virus (WNV) or several other pathogenic viruses. ... Consistent with a direct role for IFN-$\beta$ in control of WNV replication, viral titers in ex vivo cultures of macrophages, dendritic cells, fibroblasts, and cerebellar granule cell neurons, but not cortical neurons, from IFN-$\beta$(-)(/)(-) mice were greater than in wild-type cells. ... | 1 | 1 | 2 | False |

## G.4 MULTI-HOP REASONING

As discussed in Section 5.2, our token specific analysis indicates that ICR's strong multi-hop retrieval performance comes from the knowledge consolidation abilities of LLMs. In Table G.9, which shows several examples of correct in-context re-ranking by Llama-3.1 8B across three multi-hop benchmarks, it is quite evident that the **bridge entity** provides a strong signal to correctly re-rank the second hop document. It is interesting to note that in both the third and fourth examples, the final answer (both years of death) also provides additional strong signals for our method. This suggests that, in some cases, LLMs are going even farther in terms of knowledge consolidation than merely identifying the correct bridge entity.

Table G.9: In-context re-ranking scores for all tokens in several multi-hop examples from all three multi-hop benchmarks which Llama-3 8B ICR re-ranks correctly.

| Query | Bridge Entity | First Hop | Second Hop | Dataset |
|-------|---------------|-----------|------------|---------|
| Who is the mascot of the university related to Randy Conrads? | Oregon State University | Randy Conrads attended Oregon State University, graduating in 1972 with a bachelor's degree in industrial engineering. Before founding Classmates Online, Inc. Conrads worked for Boeing for twenty one years. | Benny Beaver is the official mascot of Oregon State University and winner of the 2011 Capital One Mascot of the Year write - in campaign. | MuSiQue |
| When did military instruction start at the place where Larry Alcala was educated? | University of Philippines | Larry Alcala was born on August 18, 1926 to Ernesto Alcala and Elpidia Zarate in Daraga, Albay. Through a scholarship from Manila Times granted by the publisher Ramón Roces, he obtained a degree of Bachelor of Fine Arts in Painting at the University of the Philippines (UP) in 1950. | ROTC in the Philippines began in 1912 when the Philippine Constabulary commenced with military instruction at the University of the Philippines. The university's Board of Regents then made representations to the United States Department of War through the Governor - General and received the services of a United States Army officer who took on the duties of a professor of Military Science. | MuSiQue |
| Where was the composer of film Billy Elliot born? | Stephen Warbeck | Billy Elliot Billy Elliot is a 2000 British dance drama film about a boy aspiring to be a professional dancer while dealing with the negative stereotype of the male ballet dancer. Set in County Durham, North East England during the 1984–85 coal miners' strike, it was written by Lee Hall and directed by Stephen Daldry with music composed by Stephen Warbeck. | Stephen Warbeck (born 3 May 1953) is an English composer, best known for his film and television scores. Warbeck was born in Southampton, Hampshire. | 2Wiki |
| Why did John Middleton Murry's wife die? | Katherine Mansfield | John Middleton Murry (6 August 1889 – 12 March 1957) was an English writer. He was a prolific author, producing more than 60 books and thousands of essays and reviews on literature, social issues, politics, and religion during his lifetime. A prominent critic, Murry is best remembered for his association with Katherine Mansfield, whom he married in 1918 as her second husband, for his friendship with D. H. Lawrence and T. S. | Katherine Mansfield Murry ("née" Beauchamp; 14 October 1888 – 9 January 1923) was a prominent New Zealand modernist short story writer and poet who was born and brought up in colonial New Zealand and wrote under the pen name of Katherine Mansfield. At the age of 19, she left New Zealand and settled in England, where she became a friend of writers such as D. H. Lawrence and Virginia Woolf. Mansfield was diagnosed with extrapulmonary tuberculosis in 1917; the disease claimed her life at the age of 34. | 2Wiki |
| At the 46th Grammy Awards, which award did the album by The White Stripes, which included the song Seven Nation Army, win? | Elephant | Elephant is the fourth album by the American alternative rock duo The White Stripes. Released on April 1, 2003 on V2 Records, its release garnered near unanimous critical acclaim and commercial success, garnering a nomination for Album of the Year and a win for Best Alternative Music Album at the 46th Grammy Awards in 2004, peaking at No. 6 in the US "Billboard" charts and topping the UK album charts. | Seven Nation Army" (also stylized as "7 Nation Army") is a song by American rock duo The White Stripes. It was released as the lead single from their fourth studio album, "Elephant", in March 2003, and reached number one on the Modern Rock Tracks—maintaining that position for three weeks. It also became the third best-performing song of the decade on the same chart. It was well received commercially as well, and won the Grammy Award for Best Rock Song. | HotpotQA |
| The Dukes of Hazzard was inspired by the 1975 film starring whom? | Moonrunners | The Dukes of Hazzard is an American action-comedy television series that aired on CBS from January 26, 1979, to February 8, 1985. The show aired for a total of 147 episodes spanning seven seasons. The series was inspired by the 1975 film "Moonrunners" which was also created by Gy Waldron and had many identical or similar character names and concepts. | Moonrunners is a 1975 film, starring James Mitchum, about a Southern family that runs bootleg liquor. It was reworked four years later into the popular long-running television series "The Dukes of Hazzard", and as such the two productions share many similar concepts. Mitchum had co-starred with his father, Robert Mitchum, in the similar drive-in favorite "Thunder Road" eighteen years earlier, which also focused upon moonshine-running bootleggers using fast cars to elude federal agents. "Moonrunners", a B movie, was filmed in 1973 and awaited release for over a year. Its soundtrack reflects the outlaw music boom of the 1970s during which the film was released. | HotpotQA |

