# OpenReview forum: "Attention in Large Language Models Yields Efficient Zero-Shot Re-Rankers"
_ICLR.cc/2025/Conference — ICLR 2025 Poster_

### Official Review · Reviewer_FYfW · 2024-10-29

**Soundness:** 3
**Presentation:** 3
**Contribution:** 3
**Rating:** 6
**Confidence:** 4

**Summary:**

This paper proposes an unsupervised ranking method that leverages the attention scores of query tokens over each document token. Extensive experiments, including nine single-hop and three multi-hop datasets, were conducted to prove the improvements over unsupervised baseline RankGPT. Ablation studies were conducted to prove the effectiveness of calibration and aggregation.

**Strengths:**

1. This paper is well-written and easy to follow.
2. Extensive experiments, including nine single-hop and three multi-hop datasets, were conducted to prove the improvements over unsupervised baseline RankGPT
3. Ablation studies were conducted to prove the effectiveness of calibration and aggregation.
4. Clear analyses were presented to help explain why calibration helps and what re-ranking signals the proposed method captures.

**Weaknesses:**

1) Missing important unsupervised baseline UPR:

Sachan, Devendra Singh, et al. "Improving passage retrieval with zero-shot question generation." arXiv preprint arXiv:2204.07496 (2022).

UPR ranks documents based on the log-likelihood of generating the ground-truth query given the document. Although it is less efficient than ICR, UPR operates in an unsupervised manner and should be compared with ICR.

2) The impact of document order has not been studied. ICR ranks all documents within the context, but the paper does not specify how documents are concatenated. Are they randomly shuffled before concatenation? To what extent does document order impact ranking performance?

3) For single-hop evaluations, only the weaker BM25 retriever is applied. It remains unclear whether ICR can enhance the performance of stronger retrievers.

**Questions:**

1) UPR is a popular unsupervised reranking method that ranks documents based on the log-likelihood of generating a query from the input document. How does ICR's performance compare to that of UPR?

Sachan, Devendra Singh, et al. "Improving passage retrieval with zero-shot question generation." arXiv preprint arXiv:2204.07496 (2022).

2) Are the documents randomly shuffled before concatenation? To what extent does document order impact ranking performance?

3) Why is the top 20 instead of the top 100 utilized for multi-hop evaluations?

---

> ### Author Response · Authors · 2024-11-21
> **Author response to reviewer FYfW**
>
> We sincerely appreciate the reviewer's recognition of our experiment setup, clear analyses, and writing quality. We address the concerns as follows:
>
> ## W1/Q1 Missing important unsupervised baseline UPR:
>
> We acknowledge that UPR is also a zero-shot re-ranking method and have cited UPR in the related works section. However, we argue that UPR is not a necessary baseline as it has shown to significantly underperform RankGPT 3.5 [1] and ICR (Mistral 7B) can already match the performance of RankGPT 3.5.
>
> ## W2/Q2 The impact of document order:
>
> We agree with the reviewer that the robustness to document order is an important aspect of a re-ranking method. In our paper, we did mention **ICR first reverses the document order returned by the retriever and concatenates them in the prompt (line 194-196)**, which works better than using the original order from the retriever empirically. When using randomly shuffled order, ICR has better performance than RankGPT with lower standard deviation. Please refer to the response to reviewer d1gB’s W2/Q1 for a quantitative comparison.
>
> ## W3 Can ICR enhance the performance of stronger retrievers for single-hop evaluations?
>
> Yes, ICR also works with other retrievers such as ColBERTv2. We follow existing work [1,2,3] and choose BM25 as the base retriever in the paper. To address the reviewer’s concern, we show the re-ranking performance (nDCG@10) on Trec-covid and SciFact using ColBERTv2’s top-100 results below:
>
>
>
> |     | Trec-covid | SciFact |
> | --- | :-: | :-: |
> | BM25 | 59.5 | 67.9 |
> | BM25 + ICR (Llama-3.1) | 72.8 | **76.1** |
> | ColBERTv2 | 65.7 | 67.5 |
> | ColBERTv2 + ICR (Llama-3.1) | **74.6** | 74.4 |
>
> As shown in the table, using dense retrievers such as ColBERTv2 may not always yield better performance. ICR can also improve the quality of retrieval results returned by ColBERTv2. We will add more comprehensive results in the revised paper.
>
> ## Q3 Why is the top20 instead of top100 utilized for multi-hop evaluations?
>
> So far, multi-hop retrieval has gained much less attention in the community. Our multi-hop evaluation follows earlier works [4,5] which use the documents for all questions as the retrieval corpus. The total number of supporting documents and distractors for each question in these datasets is no more than 20. The number of supporting documents are also relatively small (2-4 for MuSiQue and 2WikiMultiHopQA, and 2 for HotpotQA). Therefore, we believe re-ranking top-20 retriever outputs and reporting recall@2 and recall@5 is a reasonable experimental setup.
>
> - - -
>
> We hope our response can address the reviewer’s main concerns and respectfully request adjustment to the evaluation.
>
> [1] Sun, Weiwei, Lingyong Yan, Xinyu Ma, Shuaiqiang Wang, Pengjie Ren, Zhumin Chen, Dawei Yin, and Zhaochun Ren. "Is ChatGPT good at search? investigating large language models as re-ranking agents." arXiv preprint arXiv:2304.09542 (2023).
>
> [2] Pradeep, Ronak, Sahel Sharifymoghaddam, and Jimmy Lin. "Rankvicuna: Zero-shot listwise document reranking with open-source large language models." arXiv preprint arXiv:2309.15088 (2023).
>
> [3] Qin, Zhen, Rolf Jagerman, Kai Hui, Honglei Zhuang, Junru Wu, Le Yan, Jiaming Shen et al. "Large Language Models are Effective Text Rankers with Pairwise Ranking Prompting." In Findings of the Association for Computational Linguistics: NAACL 2024, pp. 1504-1518. 2024.
>
> [4] Gutiérrez, Bernal Jiménez, Yiheng Shu, Yu Gu, Michihiro Yasunaga, and Yu Su. "HippoRAG: Neurobiologically Inspired Long-Term Memory for Large Language Models." arXiv preprint arXiv:2405.14831 (2024).
>
> [5] Trivedi, Harsh, Niranjan Balasubramanian, Tushar Khot, and Ashish Sabharwal. "Interleaving retrieval with chain-of-thought reasoning for knowledge-intensive multi-step questions." arXiv preprint arXiv:2212.10509 (2022).

---

> > ### Comment · Reviewer_FYfW · 2024-12-02
> > **response to authors**
> >
> > Thank you for your clarification. I have decided to keep my score unchanged due to the complexity issue raised by reviewer NMvY. Instead of introducing terms like “CFP” and “DFP”, a direct comparison of complexity based on the number of floating-point operations (FLOPs) might provide a more accurate evaluation.

---

> > > ### Author Response · Authors · 2024-12-04
> > >
> > > Thank you for your engagement in the discussion. As we stated in “Response to follow-up discussion on complexity (1/2)”, since the length of input context is typically much larger than the length of output sequence for re-ranking, we can reasonably treat the computation cost (i.e. FLOPs) of a forward pass as constant and use it as a reasonable unit for comparing the complexity of re-ranking methods under the same conditions. This applies to both CFP and DFP. Therefore, considering FLOPs in the complexity analysis is equivalent to adding constant coefficients to the terms in the equations, which does not change the asymptotic complexity of algorithms. We hope this clarification can help you better understand our response.

---

### Official Review · Reviewer_CYaY · 2024-11-02

**Soundness:** 3
**Presentation:** 3
**Contribution:** 2
**Rating:** 5
**Confidence:** 5

**Summary:**

This paper proposed in-context re-ranking (ICR), an efficient re-ranking method based on the attention distributions of LLMs. This method can achieve better performance compared to RankGPT while maintaining lower latency.

**Strengths:**

The motivation is "whether autoregressive generation necessary and optimal for LLMs to perform re-ranking?", which is quite interesting and worth discussing.
The proposed method is technical sound and is effective on various datasets of re-ranking tasks.
The paper is well-written and easy to follow.
The experiments and discussions are thorough.

**Weaknesses:**

The paper lacks more high-level intuition and explanations especially about why the proposed method works intuitively.
What are the advantages and disadvantages of the proposed methods intuitively.
Why the proposed method works well although the model's instruction following ability is poor?

**Questions:**

1. In line 216, "We use N/A as a calibration query". I was wondering why use this as the special token, how about NA or None or blank space or <>? Could these be alternatives?
2. In line 226-227, "the final ranking score sd_i, which measures the change of attention weights that each document receives when the query changes from the content-free calibration query to the actual query". I was wondering whether sd_i can be considered as distance between the actual query and calibration query? It would be better to have more high-level intuition and explanation about this.
3. In line 233-234, "since we place the query tokens at the end of the re-ranking prompt, ICR can share the KV cache of document tokens when computing s_{di, Q} and s_{di, Qcal}". I was wondering why the shared KV cache is in effective when you put query tokens at end? I didn't get the point here.
4. In line 297, "ICR's performance advantage is more prominent on Mistral, which has weaker instruction-following capabilities...". I was wondering what's the advantages of the proposed method when instruction following is involved? Whether the proposed method can better leverage important information in the query? I'm looking forward to more explanation here.
5. In ablation study/discussion, I'd love to see how the overall distribution looks like in terms of calibration scores.

---

> ### Author Response · Authors · 2024-11-21
> **Author response to reviewer CYaY**
>
> We are excited that the reviewer deems our motivation "interesting" and our method "technically sound" and "effective". We present additional explanations on our intuition below:
>
> ## Weakness/Q4: lack explanations for more high-level intuition and ICR’s relationship with instruction-following.
>
> **Why does the proposed method work intuitively?**
>
> The intuition behind our method is based on the reasonable assumption that, in a well-trained LLM, more relevant documents will receive more attention when conditioned on the semantics of the query, given their strong QA and IE performance. On the other hand, when the query is content-free, the LLM should assign approximately equal attention to all documents. Therefore, we propose a calibration method to capture the intrinsic biases and remove them from the ranking score, which proves to be effective.
>
> **What are the advantages and disadvantages of the proposed method intuitively?**
>
> - Advantages: (1) ICR gets rid of the generation process which leads to high efficiency. (2) ICR can leverage various internal signals related to re-ranking, which might be overlooked in the text generation process, as detailed in Section 5.3. (3) ICR is an unsupervised and zero-shot method, making it easily applicable to any decoder-only LLMs.
>
> - Disadvantages: ICR relies on the attention pattern of the base LLM, thus its success highly depends on the quality of the base LLM. On tasks where training data is easily obtainable, ICR could underperform supervised methods.
>
>
> **Why does the proposed method work well although the model’s instruction following ability is poor?**
>
> We believe that good ‘instruction following’ ability takes two assumptions: (1) the LLM can understand the instruction and carry out required computation internally. (2) The LLM must be able to reflect the result of (1) through text generation, usually in a specialized format. Generative re-ranking methods require strong capabilities in both (1) and (2). However, ICR frees the LLM from the burden of (2) and reveals potentially hidden re-ranking capacities. This is best illustrated by the drastic improvement on Mistral 7B, whose limited generation ability results in low ranking success rate with RankGPT (usually <20%).
>
> ## Q1 Why use "N/A" as the special token?
>
> We would like to clarify that "N/A" is just a string, not a special token added to the LLM’s vocabulary. We follow Contextual Calibration [1] and use "N/A" in the paper. Empirically, we do not see a significant difference between using "N/A" and other calibration queries, such as "None", as long as they are content-free.
>
> ## Q2 High-level intuition of $s_{d_i}$, can it be considered as distance between the actual query and calibration query?
>
> We thank the reviewer for bringing up this interesting perspective. As explained in the paper, the ranking score $s_{d_i}$ measures the change of attention weights received by document $i$ conditioned on query semantics. We agree with the reviewer that $s_{d_i}$ can reflect the semantic differences between two queries based on document $i$. However, since $s_{d_i}$ is dependent on the candidate documents, it does not qualify as a distance metric for queries mathematically. Nevertheless, this looks like a very interesting direction for future research.
>
> ## Q3 Why sharing KV cache is effective when you put query tokens at the end?
>
> This is because when we put the query tokens at the end of the prompt, document token representations are not conditioned on the query and stay the same for different queries. We can therefore reuse the document tokens’ KV cache when processing different queries, instead of encoding all prompt tokens from scratch. In practice, the overhead is only 30%, rather than 100% for processing the two queries.
>
> ## Q5 How does the overall distribution look like in terms of calibration scores?
>
> We provide detailed analysis and illustration for calibration scores in Section 5.1 of the paper. We are also happy to address more specific questions regarding calibration scores.
>
> - - -
>
> We hope our response can help the reviewer better understand the intuitions behind ICR’s design and will add more explanations to the revised paper for better readability.
>
> [1] Zhao, Zihao, Eric Wallace, Shi Feng, Dan Klein, and Sameer Singh. "Calibrate before use: Improving few-shot performance of language models." In International conference on machine learning, pp. 12697-12706. PMLR, 2021.

---

> ### Comment · Reviewer_CYaY · 2024-11-28
> **Echo reviewer NMvY on cost of API call**
>
> I agree with the reviewer NMvY on the needs of the API calls. The paper claims to reduce time complexity from O(N) to O(1), it needs to be revisited and think carefully. Changing the score because of that.

---

> > ### Author Response · Authors · 2024-12-02
> >
> > Dear Reviewer CYaY,
> >
> > Thank you again for your efforts in the review and for raising your score from 3 to 5. Since today is the last day of the discussion period, please let us know if you still have any remaining concerns that we can address.

---

### Official Review · Reviewer_d1gB · 2024-11-03

**Soundness:** 2
**Presentation:** 3
**Contribution:** 3
**Rating:** 6
**Confidence:** 4

**Summary:**

This paper proposes a novel re-ranking method based on LLMs called ICR. ICR first aggregates the attention weights received by each document token from all query tokens. To mitigate intrinsic biases of LLMs, ICR calibrates the ranking scores by subtracting the attention scores obtained by using a content-free query. ICR is evaluated on both single-hop and multi_-hop re-ranking tasks using open-source LLMs, Mistral and LLaMA 3.1.

**Strengths:**

1. The problem is important, and the overall writting is clear.
2. The proposed method is easy to follow and demonstrates effectiveness on two public LLMs.

**Weaknesses:**

My primary concern is the insufficient comparison with other methods and a lack of depth in experimental analysis.

1）Although the paper compares ICR with RankGPT, highlighting improvements, it would be strengthened by comparisons with more methods, especially other zero-shot listwise methods.

[1]  RankVicuna: Zero-Shot Listwise Document Reranking with Open-Source Large Language Models.

[2] A Setwise Approach for Effective and Highly Efficient Zero-shot Ranking with Large Language Models.

2） The paper could benefit from additional analysis to deepen the understanding of ICR’s behavior. First, for open-source LLMs, it would be helpful to provide a clear comparison between ICR and fine-tuning (FT) methods, highlighting the performance gap and cases where fine-tuning might be preferable. Second, the paper does not analyze the sensitivity of the proposed algorithm to changes in the input document order. It would be valuable to examine how much the re-ranking results fluctuate when the order of input documents is adjusted.

**Questions:**

1. The analysis in Figure 4 lacks details on how results would change if the document input order were adjusted. Additionally, if the document set contains similar documents, would these similar documents receive higher weights?

2. In Section 3.3, how does the method address biases introduced by documents of varying lengths?

---

> ### Author Response · Authors · 2024-11-21
> **Author response to reviewer d1gB (1/2)**
>
> We are grateful that the reviewer finds the proposed method "novel" and "effective". Below, we address the reviewer’s concern on experimental setup and further analysis.
>
> ## W1 Although the paper compares ICR with RankGPT, highlighting improvements, it would be strengthened by comparisons with more methods, especially other zero-shot listwise methods.
>
> We appreciate the reviewer’s suggestion on comparing ICR with supervised alternatives. The focus of ICR is presenting a novel method for efficient zero-shot re-ranking using decoder-only LLMs, which is also applicable to supervised fine-tuned LLMs such as RankVicuna. Considering the sizable efficiency improvement, we believe comparing with RankGPT can already support our main claims, as other methods usually further incur additional cost in training or inference time.
>
> To resolve the reviewers’ concern, we present a comparison with supervised (RankVicuna [1]) and listwise (setwise re-ranking [2]) approaches the reviewer mentioned in the following table:
>
>
>
> |     | NFCorpus | DBPedia | SciFact | Trec-Covid |
> | --- | :-: | :-: | :-: | :-: |
> | RankGPT (Mistral 7B) | 31  | 32.9 | 65.5 | 60.6 |
> | ICR (Mistral 7B) | 33.2 | 31.4 | 72.4 | 63.9 |
> | RankGPT (Llama 3.1 8B) | 33.7 | 41.4 | 68.4 | 72.6 |
> | ICR (Llama 3.1 8B) | 34.7 | 35.3 | **76.1** | 72.8 |
> | RankVicuna | 34.7 | **44.4** | 71.3 | **80.68** |
> | Setwise.heapsort (T5-XL)* | 35.2 | 42.8 | 67.7 | 75.7 |
> | Setwise.bubblesort (T5-XL)* | **35.3** | 43.8 | 69.1 | 75.6 |
> | *RankGPT 3.5* | *32.8* | *42*  | *64.1* | *74.1* |
> | *RankGPT 4o-mini* | *38*  | *44.9* | *77*  | *79.1* |
>
> *The performance of the setwise method [2] is copied from its paper.
>
> We notice that RankVicuna and setwise methods show better performance on NFCorpus, DBPedia, and Trec-Covid, as expected. Both methods incur additional cost in training (RankVicuna) or inference time (Setwise methods). Similar to our findings in the paper, ICR shows impressive performance on SciFact, outperforming supervised methods without specialized training. We will include this comparison in the revised version for comprehensiveness.
>
>
> ## W2 A clear comparison between ICR and fine-tuning (FT) methods, highlighting the performance gap and cases where fine-tuning might be preferable:
>
> We acknowledge that supervised re-ranking methods may have stronger performance when data and compute for training is available. As discussed above, we find ICR to still excel in more challenging tasks that demand reasoning, such as SciFact.
>
> Additionally, we highlight the following advantages of ICR compared to fine-tuning methods:
>
> - **Length Generalization**: With fine-tuning methods such as RankVicuna, we usually observe a decline in performance when the number of documents increases. As noted by RankZephyr [3], fine-tuned models struggle to generalize to sliding window sizes unseen during training, even when the context window of the LLM is long enough. In comparison, by leveraging the attention patterns in general QA or IE tasks which are relatively sufficiently trained during instruction tuning, ICR can make full use of the context window of LLMs and show improved performance (Figure 3 in our manuscript).
>
> - **Ease of Use**: To keep a model up-to-date, fine-tuned methods need to retrain a new model when a stronger base model is developed or when better training data is constructed, leading to extra cost in development. In contrast, ICR is an unsupervised and inference-time method that can be applied to any new LLMs without modification. Therefore, ICR can effortlessly leverage the ever-growing capabilities of latest LLMs.
>
> We will add a discussion on supervised methods in the revised version.
>
> [1] Pradeep, Ronak, Sahel Sharifymoghaddam, and Jimmy Lin. "Rankvicuna: Zero-shot listwise document reranking with open-source large language models." arXiv preprint arXiv:2309.15088 (2023).
>
> [2] Zhuang, Shengyao, Honglei Zhuang, Bevan Koopman, and Guido Zuccon. "A setwise approach for effective and highly efficient zero-shot ranking with large language models." In Proceedings of the 47th International ACM SIGIR Conference on Research and Development in Information Retrieval, pp. 38-47. 2024.
>
> [3] Pradeep, Ronak, Sahel Sharifymoghaddam, and Jimmy Lin. "RankZephyr: Effective and Robust Zero-Shot Listwise Reranking is a Breeze!." arXiv preprint arXiv:2312.02724 (2023).

---

> > ### Author Response · Authors · 2024-11-21
> > **Author response to reviewer d1gB (2/2)**
> >
> > ## W2/Q1 Sensitivity to document input order.
> >
> > We agree with the reviewer that the robustness to document input order is essential for re-ranking methods. To address this concern, we present the re-ranking performance of RankGPT and ICR on single-hop and multi-hop datasets with different input orders. We evaluate re-ranking performance when input documents are in the order returned by the retriever (RankGPT), in reversed order of the retriever (ICR), or in random order. In the random order setting, we use 5 random seeds and report average performance as well as standard deviation. The results (nDCG@10 for Trec-Covid and recall@5 for MuSiQue) are presented in the following table:
> >
> >
> > |     | Order | Trec-Covid | MuSiQue |
> > | --- | --- | :-: | :-: |
> > | Retriever | | 59.5 | 37.9 |
> > | RankGPT | Retriever Order | **72.6** | 52.6 |
> > | ICR | Retriever Order | 66.8 | **56.8** |
> > | RankGPT | Reverse Order | **72.9** | 50.5 |
> > | ICR | Reverse Order | 72.8 | **55.9** |
> > | RankGPT | Random | 65.6 (2.58) | 49.8 (0.72) |
> > | ICR | Random | **69.8 (1.66)** | **55.6 (0.45)** |
> >
> > As shown in the table, both RankGPT and ICR work the best when the input is ranked by the retriever. We generally observe that ICR works better when using the reverse order on single-hop datasets, possibly due to the recency bias of LLMs. When the input is randomly shuffled, we notice ICR has considerably higher performance and lower standard deviation than RankGPT, indicating stronger robustness. We will add these results to the revised version.
> >
> > ## Q1 If the document set contains similar documents, would these similar documents receive higher weights?
> >
> > We observe that ICR tends to assign higher scores to documents with tokens that are similar to those in the query. In practice, we find the attention pattern of LLMs to be very sparse. The relevant tokens in similar documents receive relatively higher scores. Within these documents,  the tokens positioned at the beginning or the end of the prompt also usually get higher attention due to various intrinsic biases.
> >
> > ## Q2 How does the method address biases introduced by documents of varying lengths?
> >
> > We thank the reviewer for raising this interesting perspective.
> >
> > We assume the reviewer is referring to the possible bias introduced by longer documents receiving higher scores due to having more tokens. ICR handles this potential length bias issue via aggregation of token-level ranking scores. ICR takes the sum of attention weights received by each token in a document to compute document-level ranking scores. **Since the attention distribution of a query token sums up to 1 over the entire context window, token-level scores are not mutually independent.** Therefore, ICR’s aggregation strategy is free from the aforementioned length bias issue. In our preliminary experiments, we find this aggregation mechanism is more robust to such length bias compared to alternatives that factor in the length of a document, such as taking the average attention score which penalizes longer documents.
> > - - -
> > We hope our response could address the reviewer’s concerns and respectfully request adjustment to the initial evaluation. We also welcome further discussions.

---

> > ### Comment · Reviewer_d1gB · 2024-11-26
> > **Authors addressed concerns with experiments**
> >
> > Based on the authors' responses, additional experiments have been included to address my concerns. I have updated the recommendation score accordingly. Please incorporate these changes into the final version.

---

### Official Review · Reviewer_NMvY · 2024-11-04

**Soundness:** 3
**Presentation:** 3
**Contribution:** 2
**Rating:** 3
**Confidence:** 4

**Summary:**

The authors focus on adopting large language models (LLMs) as zero-shot rerankers. I believe this is crucial because if we also need to train the LLMs as rerankers, there would be no difference from the previous framework. The authors propose leveraging the intrinsic attention scores to aggregate into a document score, along with a standard debias score. ***The authors claim that the proposed method operates in O(1) LLM forward passes (This is a point I am skeptical about and would like the authors to clarify in their discussion).*** I will first assign a threshold score and then adjust it based on the authors' rebuttal.

**Strengths:**

1. The writing of this paper is well-crafted, allowing readers with relevant backgrounds to quickly follow and provide feedback.
2. The proposed method is versatile and applicable, making it suitable for use with open-source LLMs.

**Weaknesses:**

1. The claim of O(1) LLM forward passes raises significant questions. Please refer to my Question 1 for a detailed explanation, as this will determine the overall quality of the paper.
2. A limitation of this method is that ordinary users cannot implement it using advanced commercial large language models, especially when compared to RankGPT. However, I believe this is due to commercial factors rather than technical ones.
3. When only comparing against a single baseline, RankGPT, the classic datasets TREC 19 and 20 are notably missing.

**Questions:**

1. A listwise LLM-based ranker typically requires O(N) forward calls because it often assumes that the number of candidate documents, N, is too large, exceeding the LLM's context limit. This necessitates the use of a sliding window for multiple forward passes. Given the same number of documents, how does the proposed ICR achieve this with only two forward passes? If RankGPT requires O(N), I believe the proposed ICR would also require O(N) with the same LLM.
2. Since RankGPT is the only used baseline, the TREC 19 and 20 datasets tested in the RankGPT paper should also be included in the experiments.
3. ICR utilizes attention scores from all transformer layers. Could you provide results from an ablation study showing the outcome when using only the attention scores from the final layer?

---

> ### Author Response · Authors · 2024-11-21
> **Author response to reviewer NMvY (1/2)**
>
> We are delighted that reviewer NMvY finds our focus "crucial", our writing "well-crafted" and the proposed method "versatile". We will clarify ICR’s claim of $O(1)$ forward pass complexity and address other questions below.
>
> ## W1/Q1    ICR’s claim of $O(1)$ forward passes
>
> We would like to kindly remind reviewer NMvY that the $O(N)$ forward passes required by LLM-based generative re-ranking methods are not caused by the sliding window strategy alone. **Applying a sliding window mechanism to any listwise re-ranking method will introduce a $O(N/l)$ factor to complexity where $l$ is the stride.**  Since **the sliding window mechanism is not a necessary part of any re-ranking method**, we do not consider it when comparing complexity in the paper.
>
> Our method ICR can also be used with a sliding window strategy to handle more documents. Given the practical constraint on the context length of LLMs, we present the following complexity comparison with and without using sliding window when re-ranking $N$ documents:
>
> - Without sliding window: Let us first assume an LLM has infinite context length. An LLM generates one token in one forward pass. Consequently, for an LLM to generate a ranking list of N document identifiers, which has $O(N)$ tokens, it requires $O(N)$ forward passes. In this case, **ICR only requires two forward passes, one with the calibration query and another with the actual query, to get the ranking scores and calibration scores for all documents.** Therefore, ICR’s complexity is $O(1)$.
>
> - With sliding window: In the case of using a sliding window of size $k$ and stride $l$, a total of $(N-k)/l+1$ iterations are needed. The LLM needs to generate $O(k)$ tokens in each iteration, resulting in a total of $O(Nk/l)$ forward passes. In this case, ICR also needs $O(N/l)$ sliding window iterations and $O(1)$ forward passes in each iteration, leading to a total of $O(N/l)$ forward passes. **Therefore, ICR is still more efficient than generative methods by a factor of $O(k)$**, which is commonly set to 20 in existing literature. We argue that this is still not a trivial speed up.
>
> **In practice, we find ICR to be much more memory efficient than RankGPT** due to the absence of text generation. Therefore, ICR could support a larger window size compared to RankGPT with the same hardware. It is also worth noting that, in addition to the limited context window, an important reason that generative methods require a sliding window mechanism is that LLMs often show a decline in generative capabilities as the context length grows. This is discussed in detail in the response to reviewer d1gB’s W2.
>
> ## W2   ICR can’t be used with commercial LLMs:
>
> As the reviewer noted, “this is less of a technical issue but rather a commercial decision”. ICR is theoretically compatible with any decoder-only transformer LLMs by design. Companies who own a commercial LLM can choose to adopt ICR if they find it useful, just like what they have probably been doing for many public research ideas.

---

> > ### Author Response · Authors · 2024-11-21
> > **Author response to reviewer NMvY (2/2)**
> >
> > ## W3/Q2:   Lack experiments on TREC 19 and 20:
> >
> > We agree with the reviewer that TREC 19 and 20 are popular datasets for evaluating retrieval systems and will add these datasets to the revised paper for completeness. However, considering the small size of DL19 and DL20 (97 examples in total), we believe that the more recent and diverse BEIR benchmark can already be a strong testbed for single-hop re-ranking performance.
> >
> > We report results on DL19 and DL20 following the same settings as in the paper:
> >
> >
> >
> > |     | DL19 | DL20 | SR  |
> > | --- | --- | --- | --- |
> > |  | nDCG@10 | nDCG@10 | %   |
> > | BM25 | 50.58 | 47.96 | 100 |
> > | RankGPT (Mistral 7B) | 51.1 | 49.45 | 15.6 |
> > | ICR (Mistral 7B) | 59.17 | 53.59 | 100 |
> > | RankGPT (Llama 3.1 8B) | **66.43** | **63.6** | 99.3 |
> > | ICR (Llama 3.1 8B) | 55.65 | 51.87 | 100 |
> > | RankGPT 3.5* | 68.55 | 62.02 |  |
> > | RankGPT 4* | 75  | 70.36 |  |
> >
> > *RankGPT_3.5 and RankGPT_4 results are reported by RankGPT [1]
> >
> > We notice that ICR with both Mistral and Llama 3.1 can effectively improve retrieval performance of BM25 through re-ranking. Compared to RankGPT, we observe a significant performance improvement with Mistral 7B. ICR (Llama 3.1) surprisingly underperforms ICR (Mistral) and noticeably falls behind RankGPT (Llama 3.1). That the success rate (SR) of RankGPT (Llama 3.1) is close to 100% suggests that it is likely trained using re-ranking data, potentially even using MSMARCO, the original source of DL19 and 20, since it is a popular retrieval dataset. In addition to data leakage concerns, we also suspect such fine-tuning could lead to more skewed attention patterns which limit ICR’s performance. Upon closer inspection, we also find that BM25 retrieves many documents with high lexical overlap to the query in DL19 and DL20, making it a more challenging scenario for ICR as discussed in the paper.
> >
> > ## Q3:  Outcome when using only the last layer:
> >
> > The following table presents re-ranking performance of ICR (Llama-3.1 8B) when only using the attention weights form the last layer on single-hop (nDCG@10) and multi-hop (recall@5) datasets:
> >
> > |     | Trec-Covid | SciFact | MuSiQue |
> > | --- | --- | --- | --- |
> > | Retriever | 59.5 | 67.9 | 37.9 |
> > | ICR (Llama-3.1, last layer) | 61.8 | 67.2 | 46.8 |
> > | ICR (Llama-3.1, all layers) | **72.8** | **76.1** | **55.9** |
> >
> > We notice that only using signals from the last layer largely underperforms using all layers, indicating there are rich signals emerging from all layers of an LLM. In our preliminary experiments, we find attention patterns in the middle layers are the most informative. We also find the proposed calibration method effectively reduces the impact of different choices of layers. Since deciding which layers to use introduces extra hyper-parameters and has very limited performance gains, for the clarity of the paper, we do not further optimize for the choices of layers and leave this interesting topic to future work.
> >
> > - - -
> >
> > We hope our response has addressed the reviewer’s concerns and respectfully request a reconsideration of the evaluation.
> >
> > [1] Sun, Weiwei, Lingyong Yan, Xinyu Ma, Shuaiqiang Wang, Pengjie Ren, Zhumin Chen, Dawei Yin, and Zhaochun Ren. "Is ChatGPT Good at Search? Investigating Large Language Models as Re-Ranking Agents." In Proceedings of the 2023 Conference on Empirical Methods in Natural Language Processing, pp. 14918-14937. 2023.

---

> > > ### Comment · Reviewer_NMvY · 2024-11-26
> > >
> > > Thank you for your response. I would like to discuss the number of forward passes further.
> > >
> > > According to [1], we use the number of LLM forward passes to evaluate different LLM-based ranking algorithms. RankGPT requires O(N) LLM API calls with a sliding window approach, as stated in Table 1 of [1]. In my understanding, your work also requires O(N) LLM API calls with a sliding window approach.
> > >
> > > Of course, I understand that your method may not require generating multiple tokens. However, generating k tokens does not necessarily mean it requires k complete forward passes.
> > >
> > > [1] Qin, Zhen, et al. "Large Language Models are Effective Text Rankers with Pairwise Ranking Prompting." Findings of the Association for Computational Linguistics: NAACL 2024. 2024.

---

> > > > ### Author Response · Authors · 2024-11-27
> > > > **Author reply**
> > > >
> > > > Thank you for your reply.
> > > >
> > > > ## “In my understanding, your work also requires O(N) LLM API calls with a sliding window approach.”
> > > > We agree that when used with a sliding window strategy, ICR also requires O(N) LLM API calls, as stated in our rebuttal. However, we maintain that **the sliding window mechanism is not a necessary design for any re-ranking method**. ICR is evaluated without the sliding window mechanism throughout the paper. Therefore, we believe **it is not necessary to consider the cost brought by the sliding window mechanism when analyzing the complexity of ICR**.
> > > >
> > > > As mentioned in the response to Reviewer d1gB’s W2, RankGPT’s reliance on the sliding window strategy is caused by both (1) LLMs’ decline in generation abilities under long-context and (2) limited coverage of re-ranking training data. In fact, we observe that ICR (no sliding window) can more effectively scale from 20 documents to 100 documents than RankGPT (with sliding window) in Section 4.4 of our paper, which indicates LLMs’ attention patterns are more stable than generation quality when the context length increases.
> > > > ## “However, generating k tokens does not necessarily mean it requires k complete forward passes.”
> > > > We would like to clarify the cost of generating $k$ tokens using LLMs. In this work, we focus on large language models using the decoder-only transformer [1] architecture similar to GPT [2], which is the most popular architecture to date.
> > > >
> > > > We quote the following description of the transformer decoder [1]:
> > > > > Given $z$, the decoder then generates an output sequence $(y_1, ..., y_m)$ of symbols **one element at a time**. At each step the model is auto-regressive [3], consuming the previously generated symbols as additional input when generating the next.
> > > >
> > > > This shows that the transformer decoder indeed generates one token in one complete forward pass. Therefore, to generate $k$ tokens, a transformer decoder needs to perform $k$ forward passes. **We would appreciate it if the reviewer can specifically name more efficient inference algorithms that can generate $k$ tokens with less than $k$ forward passes.** But still, the applicability of more efficient inference methods is out of the scope of this paper.
> > > >
> > > > [1] Vaswani, Ashish, Noam M. Shazeer, Niki Parmar, Jakob Uszkoreit, Llion Jones, Aidan N. Gomez, Lukasz Kaiser and Illia Polosukhin. “Attention is All you Need.” Neural Information Processing Systems (2017).
> > > >
> > > > [2] Radford, Alec and Karthik Narasimhan. “Improving Language Understanding by Generative Pre-Training.” (2018).
> > > >
> > > > [3] Graves, Alex. “Generating Sequences With Recurrent Neural Networks.” ArXiv abs/1308.0850 (2013).

---

> > > > > ### Comment · Reviewer_NMvY · 2024-11-28
> > > > >
> > > > > ***First, generating k tokens does not necessarily mean it requires k complete forward passes, same as input different text into LLM for k times***
> > > > >
> > > > > (1) When we discuss the complexity of re-ranking methods, it's important to clarify what we mean. In re-ranking methods based on large language models (LLMs), we need to consider how to organize N documents within the LLM to determine their order. Consequently, point-wise re-ranking, pair-wise re-ranking, and list-wise re-ranking require O(N), O(N^2) (in the traditional approach), and O(N) level forward passes, respectively [1]. For instance, in pair-wise re-ranking methods, the traditional approach involves inputting all candidate pairs into LLMs to determine their rank order. This is typically considered O(N^2) complexity, although it is actually less than N^2.
> > > > >
> > > > > [1] Qin, Zhen, et al. "Large Language Models are Effective Text Rankers with Pairwise Ranking Prompting." Findings of the Association for Computational Linguistics: NAACL 2024. 2024.
> > > > >
> > > > > (2) Generating k tokens is not equivalent to re-inputting the documents k times. For example, if we input documents D1, D2, and D3 into the LLM and generate k tokens, the past computations are retained (using a KV cache) during the autoregressive generation of these tokens. This is fundamentally different from inputting different documents into the LLM three times (e.g., first time: D1, D2, D3; second time: D2, D3, D4; third time: D3, D4, D5).
> > > > >
> > > > > ***Second, it's important not to confuse reducing the number of documents (N) input to an LLM with reducing the number of tokens generated by the LLM.***
> > > > >
> > > > > (1) When it comes to the number of forward passes from N documents to the LLM, both ICR and RankGPT are essentially the same. With a sliding window approach, both ICR and RankGPT have an O(N) level forward pass. Without the sliding window, they both require only an O(1) level forward pass.
> > > > >
> > > > > (2) Therefore, from an efficiency standpoint, ***ICR gains some efficiency by not generating tokens compared to RankGPT***.
> > > > >
> > > > > (3) With this clarification, let's examine whether your main claim throughout the text holds true.
> > > > >
> > > > > Let's revisit your abstract: "Due to the absence of generation, ICR only requires two (O(1)) forward passes to re-rank documents, making it substantially more efficient than generative re-ranking methods that require at least O(N) forward passes." If you claim that ICR requires the O(1) forward pass, you are assuming the LLM can directly receive N documents. In the same situation,  RankGPT also requires an O(1) forward pass. ***It's just that RankGPT may take more time because it needs to generate multiple tokens, but it's definitely not as you claim, where you've reduced the algorithm in this field from O(N) to O(1).***
> > > > >
> > > > > ***Third, I strongly urge the AC and reviewers to engage in a discussion on this point, as I believe it is a crucial issue for this work.***

---

> > > > > > ### Author Response · Authors · 2024-11-28
> > > > > > **Response to follow-up discussion on complexity (1/2)**
> > > > > >
> > > > > > We thank the reviewer for the detailed analysis and fully agree on the importance of this discussion.
> > > > > >
> > > > > > We would like to first summarize the common ground we have reached. We agree that ICR and RankGPT share the same level of prefilling cost (input N documents to the LLM) and that “ICR gains some efficiency by not generating tokens compared to RankGPT”, as described throughout our paper and responses.
> > > > > >
> > > > > > However, we think the reviewer downplays the cost of generation required by RankGPT to “it’s just that RankGPT may take more time because it needs to generate multiple tokens.”. **In fact, the extra time required for RankGPT to “generate multiple tokens” scales at the complexity of O(N) and removing this cost is exactly where ICR contributes to more efficient re-ranking.** We will explain in detail below:
> > > > > >
> > > > > > ***First, let’s discuss the unit for measuring re-ranking complexity.***
> > > > > >
> > > > > > The reviewer states “point-wise re-ranking, pair-wise re-ranking, and list-wise re-ranking require O(N), O(N^2) (in the traditional approach), and O(N) level forward passes, respectively [1]”, which is not accurate. **The reviewer seems to confuse the concept of LLM API call with LLM forward pass.** Qin et al. [1] measure complexity in the number of API calls throughout their paper, rather than the number of forward passes, which are totally different.
> > > > > >
> > > > > > Let’s establish more concrete definition of LLM forward pass and LLM API call:
> > > > > >
> > > > > > 1. **LLM Forward Pass:** We borrow the description from Huggingface [2], which says “A language model trained for causal language modeling takes a sequence of text tokens as input and returns the probability distribution for the next token.”. This distribution is later used to sample the next output token. Therefore, a forward pass typically produces one and only one token. The cost of a forward pass is thus determined by the size of the LLM and the number of input tokens.
> > > > > >
> > > > > > 2. **LLM API Call:** Let’s refer to OpenAI’s API documentation for text generation [3], which says “OpenAI provides simple APIs to use a large language model to generate text from a prompt”. As stated in our previous reply, LLMs, as with all transformer decoders, perform this process “one element (token) at a time” [4]. This means an API call covers the process of generating a complete response given an input prompt. Therefore, the cost of an LLM API call is determined by the size of the LLM, the number of input tokens, and the number of generated tokens, which is also reflected by the pricing model most LLM vendors use.
> > > > > >
> > > > > >
> > > > > > Since the cost of an LLM API call varies by multiple factors, we argue that **the number of API calls is not a rigorous metric for comparing re-ranking complexities, although used by previous work [1]**. For example, the cost of one API call for listwise re-ranking, which generates O(N) tokens given N documents per API call, is obviously higher than pairwise re-ranking, which only generates O(1) tokens given O(1) documents per API call. Qin et al. [1] captures this discrepancy by the terms “generation AP”’ and “scoring API”.
> > > > > >
> > > > > > It is also important to note that the cost of a single forward pass increases as the sequence gets longer during generation. However, in the case of re-ranking, the number of generated tokens (N document identifiers) is typically much smaller than the number of context tokens (content of N documents). Thus, **we can roughly treat the cost of a forward pass as constant and use it as a reasonable unit for measuring complexity of re-ranking**. With this, we can compare the complexity of RankGPT and ICR measured by the number of API calls and forward passes (FP) below:
> > > > > >
> > > > > > | Approach | #API call | #FP per API call | Total FP |
> > > > > > | --- | :-: | :-: | :-: |
> > > > > > | Listwise (no sliding window) | O(1) | O(N) | O(N) |
> > > > > > | ICR (no sliding window) | O(1) | O(1) | O(1) |
> > > > > > | Listwise (sliding window) | O(N) | O(k) | O(Nk) |
> > > > > > | ICR (sliding window) | O(N) | O(1) | O(N) |
> > > > > >
> > > > > > Without a sliding window, both RankGPT and ICR require O(1) API calls. However, RankGPT needs O(N) FPs while ICR only requires O(1) FPs. With the sliding window, both methods require O(N) API calls and ICR is more efficient in terms of FPs by a factor of O(k).

---

> > > > > > > ### Author Response · Authors · 2024-11-28
> > > > > > > **Response to follow-up discussion on complexity (2/2)**
> > > > > > >
> > > > > > > ***Second, using KV-cache does not reduce decoding latency to O(1).***
> > > > > > >
> > > > > > > We first note that **using KV cache for more efficient LLM serving [5] was proposed in 2022, long after the introduction of Transformer in 2017 [4]. This work focuses on re-ranking methods instead of optimizing LLM serving. Thus it is not necessary for this work to consider KV cache for the discussion on complexity.**
> > > > > > >
> > > > > > > Even if we consider the role of KV-cache, the complexity of text generation is still not O(1), as claimed by the reviewer. To reason for this, it’s important to differentiate two types of forward passes:
> > > > > > >
> > > > > > > - One type that encodes the input prompt and generates the first token, let’s call it context FP (CFP). The KV-cache is also computed by CFP. This is used in the prefiling stage of LLM inference.
> > > > > > >
> > > > > > > - Another type that generates a subsequent token based on the input prompt and already generated tokens, let’s call them decoding FP (DFP). DFP is used in the decoding stage of LLM inference.
> > > > > > >
> > > > > > >
> > > > > > > Using KV-cache does not accelerate prefilling. While KV-cache reduces the cost of DFP to only processing one token, i.e. the previously generated token, instead of all input tokens, the generation process [2] is still autoregressive and generates one token at a time. Therefore, **using KV-cache does not reduce the complexity of generating N tokens from O(N) to O(1).**
> > > > > > >
> > > > > > > Let’s assume the context length of an LLM is long enough and discuss the case without the sliding window. In this case, the cost of a CFP is roughly the same for RankGPT and ICR. We break down the inference cost below:
> > > > > > >
> > > > > > > - The cost for RankGPT includes O(1) CFPs during prefilling (which produces the first token) and O(N) DFPs (which complete the ranking list), **leading to a total cost of O(1) CFP + O(N) DFP**.
> > > > > > >
> > > > > > > - The cost for ICR only includes O(1) CFPs, **leading to a total cost of O(1) CFP**.
> > > > > > >
> > > > > > >
> > > > > > > *If we understand correctly, the reviewer is referring to CFP in argument second.(1) of the previous reply, which agrees with our reasoning so far.*
> > > > > > >
> > > > > > > As discussed above, for re-ranking without using KV-cache, the cost of CFP and DFP are roughly the same. Therefore, ICR reduces overall complexity from O(N) FPs to O(1) FPs. When using KV-cache, both  methods need O(1) CFPs, while ICR avoids the extra O(N) DFP needed for generation.
> > > > > > >
> > > > > > > Based on the above analysis, we totally agree with the reviewer that “ICR gains some efficiency by not generating tokens compared to RankGPT”. However, we think the reviewer downplays the O(N) DFP cost required by RankGPT to “just that RankGPT may take more time because it needs to generate multiple tokens.”. **Avoiding the O(N) DFP for “generating multiple tokens” is exactly where ICR contributes to more efficient re-ranking, with or without considering KV-cache and sliding window.**
> > > > > > >
> > > > > > > So far, we have presented detailed explanations to support our claim. At the same time, if the reviewer insists that “it's definitely not as you claim, where you've reduced the algorithm in this field from O(N) to O(1)”, we would expect more concrete evidence, references, or reasoning as support.
> > > > > > >
> > > > > > > At last, we highly appreciate the reviewer’s responsible attitude toward this discussion, as it is one of the core contributions of ICR. We also respectfully ask all reviewers and the AC to carefully consider our discussion for a fair evaluation of this paper.
> > > > > > >
> > > > > > > [1] Qin, Zhen, Rolf Jagerman, Kai Hui, Honglei Zhuang, Junru Wu, Le Yan, Jiaming Shen et al. "Large language models are effective text rankers with pairwise ranking prompting." arXiv preprint arXiv:2306.17563 (2023).
> > > > > > >
> > > > > > > [2] https://huggingface.co/docs/transformers/llm_tutorial#generate-text
> > > > > > >
> > > > > > > [3] https://platform.openai.com/docs/guides/text-generation
> > > > > > >
> > > > > > > [4] Vaswani, Ashish, Noam M. Shazeer, Niki Parmar, Jakob Uszkoreit, Llion Jones, Aidan N. Gomez, Lukasz Kaiser and Illia Polosukhin. “Attention is All you Need.” Neural Information Processing Systems (2017).
> > > > > > >
> > > > > > > [5] Pope, Reiner, Sholto Douglas, Aakanksha Chowdhery, Jacob Devlin, James Bradbury, Jonathan Heek, Kefan Xiao, Shivani Agrawal, and Jeff Dean. "Efficiently scaling transformer inference." Proceedings of Machine Learning and Systems 5 (2023): 606-624.

---

> > > > > > > > ### Comment · Reviewer_FYfW · 2024-11-28
> > > > > > > > **Response to the discussion on complexity**
> > > > > > > >
> > > > > > > > Thank you to reviewer NMvY for raising the concern about the complexity and to the authors for their detailed response. Reviewer NMvY concerns about the claim about the complexity, (O(N)-->2O(1)), since there are some efficient decoding methods like KV-cache. The authors' latest response clearly presents the complexity of different scenarios. I recommend that the authors include this discussion in their paper to enhance its depth and clarity.

---

> > > > > > > > > ### Author Response · Authors · 2024-11-30
> > > > > > > > > **Discussion summary and call for engagement**
> > > > > > > > >
> > > > > > > > > We thank the reviewers who have engaged in this discussion. To streamline the process, particularly for other reviewers, the AC, and the broader audience, we provide the following summary:
> > > > > > > > >
> > > > > > > > > Reviewer NMvY raised a concern regarding ICR’s claim of reducing re-ranking complexity from O(N) forward passes to O(1), which Reviewer CYaY echoed. Assuming the LLM can process all N documents directly, we have reached agreement on two points: (1) ICR requires only O(1) forward passes, and (2) ICR achieves efficiency improvements by avoiding the generation of additional tokens. **The remaining point of debate is (3) the complexity of generative re-ranking methods, specifically the baseline method RankGPT**. Reviewer NMvY argues that the complexity of RankGPT is O(1), while we argue it should be O(N).
> > > > > > > > >
> > > > > > > > > We suspect that the disagreement on point (3) arises from potential confusion between an LLM API call and a forward pass, possibly influenced by the widely used KV-cache technique in LLM inference. Using KV-cache accelerates decoding forward passes, but does not change the complexity of the decoding process. To address this, **we provided a detailed analysis in our follow-up response, demonstrating that the complexity of RankGPT is indeed O(N) forward passes, thereby supporting our main claim**. This analysis was acknowledged by Reviewer FYfW as “clear”.
> > > > > > > > >
> > > > > > > > > Since this appears to be the sole remaining concern raised by reviewers for this paper, we respectfully ask other reviewers to engage further in the discussion and help reach a consensus.

---

> > > > > > > > > > ### Author Response · Authors · 2024-12-02
> > > > > > > > > >
> > > > > > > > > > Dear Reviewer NMvY,
> > > > > > > > > >
> > > > > > > > > > Thank you again for your valuable comments and for initiating this important discussion. Since the discussion period concludes today, we kindly ask you to review our latest response regarding the complexity of ICR and RankGPT and see if it addresses your concern.

---

### Author Response · Authors · 2024-11-25
**Submission Updated By Authors**

We thank all reviewers for their insightful comments and have revised our submission accordingly. In addition to minor proof-reading, we summarize the major changes below (highlighted in blue in the revision):
1. We added an explanation to our intuition behind ICR’s design in the abstract (reviewer CYaY).
2. We enriched explanations of ICR’s design in Section 3 for better readability (reviewer CYaY and d1gB).
3. For Section 4, we enriched single-hop experiment results (table 1) with results on DL19 and DL20, which enhances the comprehensiveness of our evaluation setup (reviewer NMvY).
4. For Section 5.1, we added evaluations and analysis on the sensitivity of ICR to different input order to Appendix C, which shows ICR is more robust than RankGPT to different input orders (reviewer d1gB and FYfW).
5. We added re-ranking results based on ColBERTv2 to Appendix D, which shows that ICR also works with stronger base retrievers (reviewer FyfW).
6. We added comparison between ICR, RankVicuna, and Setwise re-ranking to Appendix E along with relevant discussions (reviewer d1gB).

We hope our responses and the submission update can resolve the concerns raised by the reviewers.

**As the discussion period approaches its end, we kindly ask the reviewers to read our response and adjust their evaluations. We remain open to addressing any further concerns.**

---

### Author Response · Authors · 2024-12-04
**Summary of the author response period and additional clarification on complexity**

We're glad that the reviewers acknowledged several strengths of this work, specifically the good writing quality (all reviewers), ICR's novelty and interesting motivation (Reviewers d1gB and CYaY), effectiveness (Reviewers d1gB and CYaY) and versatility (Reviewer NMvY). Additionally, Reviewers CYaY and FYfW think our experiments and analysis were thorough.

In the author response period, we have addressed most concerns from reviewers:
- Reviewer NMvY mainly raised concerns on the validity of our claim on complexity improvement and the lack of experiments on TREC 19 and 20. We have updated our draft with additional experiments on the two datasets and provided detailed explanations for complexity to address these concerns.
- Reviewer d1gB asked for comparisons with more methods and evaluation on ICR's sensitivity to different document order. We responded with a comparison between ICR and two methods named by the reviewer, as well as evaluation of ICR's robustness to varying document orders, which addressed the reviewer's concerns.
- Reviewer CYaY was mainly concerned with the lack of explanations for the intuition behind ICR. We provided detailed explanations in our response and updated the paper draft accordingly.
- Reviewer FYfW raised concerns about one missing baseline method, ICR's robustness to document order, and its applicability to stronger retrievers. We responded with justification of our baseline choice. We also demonstrated ICR's robustness and effectiveness with stronger retrievers through additional experiments.

No new concerns have been raised during the author response period. The only remaining point of concern is whether ICR reduces the complexity of LLM-based zero-shot re-ranking from O(N) forward passes to O(1), as we claimed. This was first raised by Reviewer NMvY and echoed by Reviewers CYaY and FYfW. We have already provided detailed discussion on this matter and would like to add to our previous discussion summary with some final clarifications.

To our understanding, Reviewer NMvY’s concern arises from two aspects:

- Firstly, the reviewer thinks that ICR also requires a sliding window strategy to handle the same amount of documents as RankGPT, thus ICR also needs an O(N) number of forward passes.

    We have responded to this aspect in our rebuttal. We acknowledge that using ICR assumes the LLM has enough context length to process all documents. It is important to note that ICR’s design only relies on the attention pattern of LLMs, which is orthogonal to the efforts in improving context length and inference efficiency of LLMs. We believe the increase of LLM context length and the advancement of inference infrastructures will continue to weaken the limitation imposed by this assumption.
- Secondly, the reviewer thinks RankGPT also only needs O(1) forward passes if used without the sliding window mechanism.

    We have provided detailed explanations in our discussions and showed this is not true. RankGPT still requires O(N) forward passes in this case.

We hope our explanations can help the reviewers and other audience to clear potential confusion and resolve this concern.

---

### Meta-Review · Area_Chair_2U9B · 2024-12-20

**Metareview:**

The paper proposes an approach to zero-shot reranking for information retrieval. The idea is to measure how much attention is placed on a particular document, relative to some dummy calibration query. Notably, this approach does not require any auto-regressive decoding, unlike some prior works. Consequently, for re-ranking large document sets, the approach offers a sizable latency reduction, while possessing good quality.

Reviewer opinion was divided. On the one hand, the paper studies a topical problem, and has an interesting solution which is intuitive. The empirical results are encouraging, and showcase the flexibility of LLMs for complex tasks.

On the other hand, there were concerns about the paper's claims about reducing $O(N)$ to $O(1)$ forward passes for the re-ranking. Following a lengthy discussion, some salient points emerged:
- there is a distinction between "forward passes" and "API calls", with the latter favoured in some prior work.
- "forward passes" can encompass both a prefill/context encoding step, and sequential auto-regressive decoding steps. These steps have rather different profiles; e.g., the former tends to be compute-bound.
- for both the proposed method and RankGPT, the input is a (tokenised) query and list of (tokenised) documents.
- in RankGPT, one performs a single parallel prefill/context encoding step; and then $N$ sequential auto-regressive decoding steps, to generate the proposed ranking over the documents.
- in the proposed method, one performs two parallel prefill/context encoding steps; and _no_ auto-regressive decoding steps.

The paper's claim is then that the proposed method can be much faster since it has shaved off $O(N)$ auto-regressive decoding steps. This claim was not explained in detail in the original manuscript (which simply referred to "forward passes", without discussing context encoding versus auto-regressive decoding). We believe it crucial for the paper to properly discuss this issue, and ensure there is precision in the claims. This said, we believe it accurate that the proposal does indeed shave off $O(N)$ auto-regressive decoding steps, and that this can lead to significant cost reductions.

There were also some questions regarding the paper's comparison to other baselines (e.g., listwise ranking), stability analysis, and providing more intuition for the method. The author response on these points were reasonable, and were generally seen as satisfactory.

Overall, the paper's finding is expected to be of interest to the retrieval community. It is however imperative that the comments about "forward passes" are properly qualified. We strongly encourage the authors to incorporate detailed points made in the discussion with reviewers.

**Additional Comments On Reviewer Discussion:**

There was a detailed back-and-forth on several points, but most notably the one about "forward passes". Following this exchange, there was clarity on the distinction between prefill and decoding steps, with the paper's claim being a reduction in "forward passes" during the _decoding_ step only. Per above, the paper would benefit from including such discussion in an updated version.

---

### Decision · Program_Chairs · 2025-01-22

Accept (Poster)